

# Calibration of the 2007-2017 record of ARM Cloud Radar Observations using CloudSat

Pavlos Kollias[1,2], Bernat Puigdomènech Treserras[3] and Alain Protat[4]

[1]School of Marine and Atmospheric Sciences, Stony Brook University, USA

[2]Center for Multiscale Applied Sensing, Brookhaven National Laboratory, USA

[3]Department of Atmospheric and Oceanic Sciences, McGill University, Canada

[4]Bureau of Meteorology, Melbourne Australia

*Correspondence to*: Pavlos Kollias (pavlos.kollias@stonybrook.edu)

**Abstract.** The US Department of Energy (DOE) Atmospheric Radiation Measurements (ARM) program has been at the forefront of millimeter wavelength radar development and operations since the late 90's. The operational performance of the ARM cloud radar network is very high; however, the calibration of the historical record is not well established. Here, we use a well-characterized spaceborne 94-GHz cloud profiling radar (CloudSat) to characterize the calibration of the different

generations of the ARM cloud radars from 2007 to 2017 over a variety of climatological regimes and for fixed and mobile deployments. Over 43 years of ARM profiling cloud radar observations are compared to CloudSat and the calibration offsets are reported as a function of time using a sliding window of 6 months. The study also provides the calibration offsets for each operating mode of the ARM cloud radars. Overall, significant calibration offsets are found that exceed the uncertainty of the technique (1-2 dB). The findings of this study are critical to past, on-going and planned studies of cloud and precipitation and

should assist the DOE ARM to build a legacy decadal ground-based cloud radar dataset for global climate model validation.

## 1 Introduction

The first millimeter wavelength cloud radars (MMCR, [Moran et al., 1998]) of the U.S. Department of Energy Atmospheric Radiation Measurement (ARM) program were installed at the Tropical Western Pacific (TWP) Manus and Southern Great Plains (SGP) sites in 1996. Since then, the ARM program has been at the frontier of short-wavelength radar development and

25 operations for over two decades [Kollias et al., 2016]. At the beginning, emphasis was placed on demonstrating high operational stability and in developing standard hydrometeor location and spectral products ([Clothiaux et al., 2001] [Kollias et al., 2007b]). The ARM program MMCR calibration efforts were limited to subcomponent characterization (i.e. antenna gain), monitoring of the transmitted peak power and infrequent detail characterization of the radar receiver by injecting signal with known amplitude. In 2005, the ARM program started the deployment of its mobile facilities and the gradual modernization

of the MMCR receiver. This led to the development of the W-band ARM Cloud Radar (WACR). In 2009, the ARM program embarked in a significant expansion of its radar facilities [Mather and Voyles, 2013]. The expansion included the addition of



scanning mm- and cm-wavelength radars with Doppler and polarimetric capabilities ([Kollias et al., 2014a], [North et al., 2017]) and the development of the next generation profiling cloud radar, the Ka-band ARM Zenith-pointing Radar (KAZR) and its upgraded, second generation (KAZR2).

The strengthening of the microphysical retrievals is one of the main drivers for routinely operating more than one radar frequencies in the atmospheric column. As expected, this requirement brought the characterization of the ARM radar calibration to focus. Early comparisons between collocated profiling ARM cloud radar indicated differences in reported radar reflectivity profiles. This hardly came as a surprised to those involved in radar characterization [Atlas, 2002]. Soon after the National Aeronautics and Space Administration (NASA) Tropical Rainfall Measuring Mission (TRMM) spaceborne radar was

on orbit, its remarkable stability made it a calibration standard and its comparison to the ground-based observations of the Weather Surveillance Radar -1998 Doppler (WSR-88D) network uncovered several issues with the calibration of the radars [Bolen and Chandrasekar, 2000]. Arguably, calibrating a small network of profiling cloud radars is a far more challenging task. The systems are only vertically pointing, thus, makes the use of corner reflectors or metal spheres difficult; designed with sensitive receivers that can detect very low radar reflectivity targets but saturate in rain, thus, making the use of disdrometers

challenging ([Gage et al., 2000]); operate in climate regimes that often have no or little precipitation and suffer by considerable gaseous and hydrometeor attenuation ([Kollias et al., 2005]; [Kollias et al., 2007a]). Furthermore, the four different profiling cloud radars (MMCR, WACR, KAZR, and KAZR2) were deployed in several different climatological regimes, for small periods of time (9-24 months mobile deployments) and often with no gaps between deployments, thus, making it even more challenging to develop calibration standards. At present, the ARM program employs a larger radar operations and engineering

group and have set procedures for characterizing the ARM radars using a combination of subsystems calibration, corner reflectors and natural targets. However, these methods are still not fully operational today and certainly not applicable to the historic ARM profiling cloud radar dataset that spans over two decades.

Luckily, NASA's CloudSat mission, a 94-GHz spaceborne Cloud Profiling Radar (CPR) was launched on April 2006 ([Tanelli

et al., 2008]) on a circular sun-synchronous polar orbit providing coverage from 82º S to 82º N and continues to operate to present. In 2021, another 94-GHz spaceborne CPR with Doppler capability will be launched as part of the Earth Clouds, Aerosols and Radiation Explorer (EarthCARE) satellite a joint European Space Agency and Japanese Aerospace Exploration Agency mission ([Illingworth et al., 2015]; [Kollias et al., 2014b]). Over the 12-year mission of CloudSat, end-to-end system calibration is performed using measured backscatter off the ocean surface and the calibration of the CloudSat reflectivity

measurements is accurate within 0.5 - 1 dB ([Li et al., 2005]; [Tanelli et al., 2008]). The CPR calibration quality and stability was exploited by [Protat et al., 2011] who, first demonstrated, that using a statistical approach, CloudSat could be used as a global radar calibrator for ground-based profiling cloud radars. In the Protat et al., 2011 study, two ground-based radars, the MMCR at the North Slope of Alaska (NSA) Barrow ARM site and another 35-GHz radar system at Cabauw, The Netherlands were calibrated using CloudSat over a short period of time (6-12 months).



Here, the Protat et al. [2011] methodology is revised, improved and applied to almost the entire record of ARM profiling cloud radar observations at the fixed and mobile sites from 2007 to the end of 2017 (at total of 43.5 years of radar observations). The application of the technique is such diverse set of radar systems and locations is expected to demonstrate the applicability of this approach to existing profiling radar networks such as the ARM program and the future European research infrastructure

5     network for the observations of Aerosol, Clouds and Trace gases (ACTRIS). The characterization of a decade long cloud radar record from multiple locations is a necessary step for the development of unbiased statistics on cloud occurrence and the estimation of microphysical parameters using retrieval techniques. Once the characterization and reprocessing of the ARM radar observations is completed, the decade long record and its value-added products can be used as an observational targets for Global Climate Model evaluation studies using suitable forward operators ([Zhang et al., 2018], [Lamer et al., 2018]).

## 2 Methodology

Here, the ARM and CloudSat CPR measurements and the methodology used for the comparison between the ground-based and space-based observations are described

### 2.1 ARM Cloud Radar Measurements

The record of ARM profiling radar observations compared to CloudSat is detailed in Table 1. A total of 14 different locations (Fig. 1) with four different radar systems (MMCR, WACR, KAZR, KAZR2) for a total of 43 years and 8 months long record of radar observations. At couple sites, the calibration record starts as early as the launch of CloudSat (mid 2006) and in several

sites stops at the end of 2017. For much of the record analyzed here, the WACR was the primary profiling cloud radar of the first ARM Mobile Facility (AMF) and as such have been deployed in several different climatological locations. A marine version of the WACR (M-WACR) with smaller antenna and on a ship-motion stabilizer has been the primary radar for marine deployments of the second AMF (AMF2). The WACR does not use pulse compression and operates only in co-polarization and cross-polarization modes. The single operating mode of the WACR combined with the fact that uses the same frequency

as the CloudSat CPR makes their comparison relatively straightforward. The MMCR used a complicated operating mode sequence ([Moran et al., 1998]; [Kollias et al., 2007b]) in order to meet the requirement of detecting all radiatively important clouds with radar reflectivity above -50 dBZ throughout the troposphere. The mode sequence includes a long pulse compression mode for detecting high level clouds (hereafter Mode 2), a very short pulse for boundary layer clouds detection, a nominal length general mode that covers all the troposphere (hereafter Mode 3) and a precipitation mode that provides

additional receiver protection to avoid signal saturation. These modes operated in a interleave sequence. The KAZR system provides the chirp (hereafter mode MD) and general mode (hereafter mode GE) on the same time using a dual radar receiver channel with enough frequency separation to enable detection of two pulses transmitted on the same trigger. Finally, the KAZR2 is an improved hardware version of the KAZR, maintains the same operating modes as the KAZR but introduces also a precipitation mode that transmits a reduce amplitude pulse to avoid receiver saturation by strong precipitation returns.



The use of different modes by the majority of the ARM profiling cloud radars allow us to achieve high sensitivity for all cloud system and enables us to avoid velocity aliasing and receiver saturation effects. However, the use of different modes comes at the expense of frequent range sidelobe artifacts from high reflectivity targets from the use of pulse compression and possible

differences in the reported radar reflectivity from the different modes. The latter is commonly observed in radar systems that operate with different modes. While the process of tracking down the detail hardware and/or software issue responsible for the intramode differences is difficult, their temporal evolution can provide clues for identifying time periods where the performance of the radar hardware changed or when the radar data post-processing algorithms were changed. A detailed comparison between the reported radar reflectivities from all the radar systems with more than one operating mode was

conducted (Fig. 2). The difference between mode 3 and mode 2 is reported for the MMCR systems and the difference between the GE and MD modes is reported for the KAZR and KAZR2 systems. The difference [dB] in the measured radar reflectivity between two modes is estimated at ranges both modes provide observations (e.g., the MMCR mode 2 does not provide data below 3.6 km) and at ranges where the averaged profiles were correlated to filter our ranges where bid discrepancies due to radar artifacts were present. At each height, the average reflectivity profile of each mode (in linear units) is computed. The

mean of the differences in the averaged radar reflectivity profiles between the two modes is computer and shown as a function time in Fig. 2. Overall, the differences are small (±2 dB) and only in few MMCR cases, the differences are much high, suggesting that during these periods, the hardware performance was questionable or that the radar operators performed tests. Overall, few changes in the relative difference between the modes in observed in the next generation ARM cloud profiling radars (KAZR and KAZR2). These differences are small, and it is difficult to identify which of the two radar reflectivities is

correct unless an independent radar calibration procedure is available.

## 2.2 The ARM – CloudSat comparison methodology

The comparisons between the ARM radars (MMCR, KAZR and KAZR2) and the CloudSat CPR are performed independently for two modes for the MMCR (2 and 3) and two modes for the KAZR and KAZR2 (MD and GE). The approach is similar to

25 that described in [Protat et al., 2011]. The technique consists of a statistical comparison of the mean vertical profiles of non-precipitating ice cloud radar reflectivities from the ground-based and spaceborne radar observations. One of the improvements introduced in this study is that the averaging of the radar reflectivity value at each height in performed in linear space (Z) and not dBZ as in [Protat et al., 2011]. These averaged profiles use data extracted from CloudSat overpasses in a radius of 200 km around the ARM site and ±1 h time lag around the overpass time for the ground-based radars. Another improvement introduced

in this study is a rigorous selection of the CloudSat overpasses within a certain radius to avoid any errors in the estimation of the proximity of CloudSat columns to the ARM site location. Finally, the methodology investigates the difference between the ARM and CloudSat profiles in a large range of calibration offsets from -15 to +15 dB with a fine spacing of 0.1 dB rather than using an iterative process as in [Protat et al., 2011]. The radar reflectivity difference between the ARM and CloudSat profiles is evaluated only at the range of heights where enough samples from both sensors are available.





Several factors need to be taken into account to achieve an objective, statistical comparison between ground-based and space-based observations: frequency of each radar, sensitivity, viewing geometry, attenuation correction, etc. The approximations to deal with all these factors introduce sources of error that are difficult to assess. A detailed description of all necessary steps required to find the calibration offset for each radar are described here, following the algorithm flow outlined in Fig 2.

The CloudSat overpasses are predicted using the two-line element set (TLE) files that encode all necessary information to define the latitude and longitude location of the satellite over the Earth's surface at a given time. Using these files, the satellite position is computed with high resolution in time and the distances to each ARM radar location are used to define the overpass. Only CloudSat data passing in a radius between 200 and 300 km around the ARM radar location are extracted. Knowing the orbits of the overpasses, the CloudSat respective files are read. In this present study, the data from the fourth and fifth release (R04 and R05) of the 2B-GEOPROF product are used to extract the CPR radar reflectivity, height, DEM elevation, CPR cloud mask, gaseous attenuation and data quality flags. In addition, the height of the freezing level is extracted from the 2C-PRECIP-COLUMN product. Fig 4a shows the pdf of the freezing level height at the NSA.

All CPR observations within 500 m from Earth's surface are removed to avoid residual surface clutter contamination. In addition, all CPR detections at very low Signal-to-Ratio (SNR) conditions (CPR Cloud Mask < 20) and poor data quality points (Data Quality ≠ 0) are removed.

The gaseous attenuation correction reported in the CloudSat files is added to the reflectivity profile. The CPR reflectivity is normalized for the differences in the values used for the dielectric constant (K) using Eq. 1. CloudSat uses a value of 0.75 and the ARM program uses a value of 0.99 for all MMCR, 0.84 for all WACR and 0.88 for all KAZR.

$$Z_{CloudSat} = Z_{CloudSat} - 10 \cdot \log_{10}\left(\frac{K_{ARM}}{K_{CloudSat}}\right) \qquad (1)$$

On the ARM radar data processing, only data with ±1h time lag around the overpass exact time is used. All radar reflectivity values measured at SNR < - 15 dB conditions are removed. The ARM radar reflectivities are corrected for gaseous attenuation using the top-down gaseous attenuation profile $(G_{94-GHz})$ available in the CloudSat data products. First, the profile in inverted $\left(G_{94-GHz}^{inv}\right)$ to represent the gaseous attenuation for a ground-based system. If the ground-based system is the WACR (same frequency with CloudSat), no further conversion is needed and the $G_{CloudSat}^{inv}$ is used to correct the WACR radar reflectivities. If the ground-based system is a 35-GHz radar (MMCR, KAZR, KAZR2), then a conversion factor C is used to convert the $G_{94-GHz}^{inv}$ to 35-GHz gaseous attenuation $(G_{35-GHz})$ using Eq. 2.





$$G_{35-GHz} = \frac{G^{inv}_{94-GHz}}{C} \qquad (2)$$

The conversion factor is derived using [Rosenkranz, 1998] and its average value depends on the ARM radar location (i.e., 1.45 for AWR, 2.08 for NSA, OLI and TMP, 3.36 for ENA and SGP and 4.03 for TWP) . If the ARM cloud radar operates at 35-GHz, another important step is to address the difference in the scattering of ice particles at 35- and 94-GHz. Here, we use the relationship introduce by [Protat et al., 2010], that is applied for reflectivity values lower than 30 dBZ and it is shown in Eq. 3:

$$dBZ_{94GHz} = dBZ_{35GHz} - 10^{-16.8251} \cdot (dBZ_{35GHz} + 100)^{8.4923} \qquad (3)$$

Subsequently, the ARM radar reflectivities are averaged to 1-min using linear averaging and 250 m vertical resolution to best match the CloudSat footprint (~1.4 km) and range resolution. If there are data from both radars for a given overpass, the following processing prepares the data for the final statistical comparison:

1) The profiles are carefully separated in 2 groups: precipitating and non-precipitating ice clouds. Ice clouds are assumed at heights above the freezing level while liquid particles are assumed below. An ARM column is considered to be precipitating if 10% or more of the if at least 10% heights below the freezing level report hydrometeor echoes. For CloudSat columns a maximum of 35% of the heights below the freezing level are allowed to report hydrometeor echoes before the column is characterized as precipitating. Precipitating profiles are eliminated from the ARM-CloudSat comparison since they are not attenuated the same way from nadir or zenith viewing geometries. This conservative selection will ensure that only non-precipitating ice clouds observations, that have negligible hydrometeor attenuation are used.

2) The performance of the most sensitive radar is degraded to match the minimum detectable signal (MDS) of the less sensitive radar. Due to the large distance between CloudSat and the Earth's troposphere, the CloudSat MDS is practically constant around -30 dBZ throughout the troposphere, while the ARM radar MDS decreases with the square of the range from the radar.

3) If the ARM radar operates at 35-GHz, the radar reflectivity is converted to 94-GHz radar reflectivity using Eq. 3.

4) Using all available column within the selected time window (6 months), a reflectivity frequency by altitude diagram (CFAD) is constructed for each radar (Fig. 4b,c). This diagram will be used to generate the mean vertical reflectivity profile used in the final comparison (Fig. 4e).

5) Steps 1 and 3 are repeated for all possible calibration offsets, from -15 to +15 dBZ with increments of 0.1 dBZ. At each iteration, the calibration offset is added to the original profile prior to the frequency conversion (prior to step 3) and 1 CFAD is constructed for each calibration offset accumulating columns from all overpasses.



6) Each CFAD constructed with the previous methodology is representative of one averaged profile. As we have N calibration offsets, we have N averaged profiles for each CloudSat and ARM radar (Fig 4e).

7) The final calibration result is found by computing the root mean square error (RMSE) between the profiles of each radar for each calibration and at heights with enough data points. The calibration offset representative of the profiles with the least RMSE will be the final calibration result (Fig. 4f).

8) The probability density function (pdf) of cloud top heights (Fig. 4d) is also used for verification purposes, assuming that occurrences of the highest clouds should be similar when the ground and spaceborne radars have equal sensitivity, [Protat et al., 2010].

The most important factor in determining our ability to perform a good comparison is the number of available CloudSat profiles. Several temporal windows were considered, and the decision was made to use a time window of 6 months throughout this study. In addition to the length of the time window, the impact of the maximum distance of the CloudSat observations from the ARM site (we tested values from 100 to 300 km) and of the maximum fraction of hydrometeor echoes allowed in warm temperatures was investigated. Fig. 5 shows the number of CloudSat profiles with suitable measurements (non-precipitating ice) with a 6-month window for all the ARM fixed and mobile sites as a function of time. As expected, there is strong seasonal variability that is dictated by the seasonal cloud type and atmospheric temperature profile variability. Of particular interest is the availability of suitable CloudSat profiles at the NSA. There is a significant decrease in the number of available CloudSat profiles during the period that the ARM program transit from the MMCR to the KAZR radar system. The reduction in the number of CloudSat profiles is not related to the changes of the ARM radar system (these two systems have similar MDS) nor is related to significant changes in the cloud climatology at the NSA. The transition from the MMCR to the KAZR system coincided with the battery anomaly occurred on CloudSat in 2011 that resulted in CloudSat operating since then only during daylight conditions, thus, effectively halving the possible number of CloudSat columns ([Stephens et al., 2018]). The daylight-only operations of CloudSat challenged our ability to collect a good size sample of column especially at very high latitudes (e.g. AWR during the southern hemisphere winter).

A total of 653 ARM – CloudSat comparisons were performed using a running 6-month time window. The relationship between the minimum RMSE value achieved in a particular ARM – CloudSat comparison and the corresponding number of CloudSat columns is shown in Fig. 6. As expected, the RMSE value reduces with the number of samples. The analysis of the entire ARM-CloudSat comparison record suggests that when the number of CloudSat columns is less than 500 the comparison is difficult to perform. In addition to the value of RMSE and the number of CloudSat columns, the goodness of the fit between the ARM and CloudSat cloud top height pdf's is evaluated when the minimum RMSE is achieved. Out of the possible 653 calibration coefficients, 616 were accepted, a 94.3% success rate.



## 3 Results

First, the results of the ARM – CloudSat comparison at the two sites that feature the most recently acquired profiling cloud
radar systems of the ARM program are discussed. The two KAZR2 systems are located at critical climatological locations
(ENA and OLI) and are the primary source of cloud observations. The OLI KAZR2 is compared against the CloudSat CPR
for the period 09/2015 to 12/2017. Fig. 7a shows the calibration offset (dB) we need to add to the MD mode observation to
minimize the RMSE with the CloudSat observations. If the calibration offset is positive, this suggests that the MD mode
underestimates the radar reflectivity compare to CloudSat.  Although a 6-month running time window is used, considerable
temporal variability is observed especially at the beginning of the period.  At the beginning of the period, -2.3 dB need to be
added to the ARM observations to statistically minimize their differences against the CloudSat observations. During the first
4 months of 2016, + 3.4 to + 4.6 dB need to be added. The last estimate of this 4-month period is higher (+6.9 dB) and coincides
with a period when considerable changes occurs in the radar hardware/software and the calibration offset is back to -2.3 dB.
Through our analysis, every time the ARM radar hardware and/or software (including receiver signal processing) undergo a
change, we noticed that the ARM-CloudSat comparison where challenging to achieve. This is attributed to the fact that part of
the 6-month observing period uses observations with one configuration and the other part use observations with a different
configuration. After this period, the calibration offset changes slowly raising to +1-3 dB in early 2017 and during the latter
part of 2017 the calibration offset is less than +0.5 dB.

Fig. 7b shows the calibration offset for both KAZR2 operating modes (MD and GE) using ARM – CloudSat comparison
methodology applied to the recorded radar reflectivities of each mode. Overall, the calibration offsets closely follow each other
throughout the observing period. During the first six month, the calibration offset for the MD is about 1 dB higher, suggesting
that the MD reports on average 1-2 dB lower radar reflectivities than the GE mode. This relationship is reversed around
04/2016 and until the end of the observing period, the calibration offset for the MD mode is now 1-2 dB lower than that
estimated for the GE mode. to match the CloudSat observations.  Noticeably, the reversal in relationship of the calibration
offsets coincides with the period we argued earlier that coincides with changes in the radar configuration around 04/2016.
During that period, the number of FFT in the recorded radar Doppler spectra changed from 256 to 512 and the calibration was
updated (Joseph Hardin, ARM radar engineer, personal communication).

As discussed in section 2.1 the ARM MD and GE modes observations can be used to estimate their relative offset. Fig. 7c
shows the difference (MD – GE) in dB of the two KAZR2 operating modes (black line). On the same plot, the difference (MD
– GE) is dB as seen from CloudSat is also reported (circles).  Overall, very good agreement is found between the two estimates
of the radar reflectivity offset between the two KAZR2 modes. This suggest that the ARM – CloudSat comparison can provide
high quality information regarding the absolute and relative calibration offsets between radar modes.
The second KAZR2 system is operated at the ENA since the fall of 2015. Fig. 8 shows two calibration offset (dB) values for
the KAZR 2 MD (white symbols). Contrary to the OLI site, the ENA site cloud and temperature climatology do not favor the
collection of large number of suitable CloudSat columns for calibration (Fig. 5). During the first 9 months of operation
(10/2015 -07/2016) the calibration offset was very small (+0.3 dB) indicating that the radar was well calibrated. During the



last 10 months of the observing period (01/2017 – 10/2017), the calibration offset is +5.2 dB. In an attempt to independently verify the observed trend in the KAZR2 calibration offset, the Parsivel disdrometer particle size distribution (PSD) measurements available at 1-min temporal resolution are used. The difference between the Parsivel-derived radar reflectivity and the KAZR2 radar reflectivity is shown on Fig. 8 (white dotted line) and suggests a trend, similar to the calibration offset

estimated from the ARM - CloudSat comparison. Additional information regarding the estimation of the KAZR2 calibration offset using the Parsivel disdrometer can be found in Appendix A.

The ARM TWP Darwin, Manus and Nauru sites are located deep in the tropics and featured MMCR system until the first quarter of 2011. Only at two sites (Darwin and Manus) the MMCR systems were replaced by KAZR systems. All TWP sites

terminate operations in 2014 ([Long et al., 2016]). The calibration offsets for the period 2007 to 2014 at the TWP sites are shown in Fig. 9. The calibration offsets record is not continuous since the number of CloudSat columns is affected by the significant inter- and intraseasonal cloud and precipitation variability driven by large scale features at different temporal-spatial scales such as El-Nino Southern Oscillation, the Madden-Julian Oscillation, and the movement of the intertropical convergence zone (ITCZ). The operational record of the TWP systems is also intermittent due to the logistical challenges associated with

the physical presence of ARM engineers at these sites; delays associated with the delivery of hardware components at the TWP sites, and poor communications for instrument monitoring especially at Manus and Nauru ([Long et al., 2016]). Overall, the calibration offsets are within ± 6 dB. The ARM intramode differences in the reported radar reflectivities are also reported (gray circles) to help interpreter the estimated calibration offset trends. The Darwin MMCR exhibits the highest variability in the ARM intramode differences suggesting frequent intentional or unintentional changes in the MMCR hardware and/or software.

During these periods, no reliable ARM-CloudSat calibration offsets are estimated. The Darwin KAZR GE calibration offset record is very sparse due to long periods with no observations. Noticeably, on GE mode observations are available at the Darwin and Manus sites. At Manus, the ARM intramode differences are small (less than 1 dB) and remain stable over a long period (3.5 years). As a result, we have calibration offset estimates for the entire observed period. The Manus MMCR 2 calibration offset gradually drifts from negative in 2007 to near zero for almost all 2008, then raises to positive +7 dB in early

2009 and after the middle of 2009 to the end of its observational record slowly fluctuates ± 3 dB. The KAZR GE calibration record is also sparse with a small calibration offset during its early operation and a +5 dB offset during the late period of its operational record at Manus. Finally, the record of MMCR observations at Nauru that overlaps with CloudSat operations in space is short (1.5 year). During that period, the ARM intramode differences fluctuate between two stages (+1.5 dB and near 0 dB). The ARM – CloudSat calibration offsets also fluctuate temporally in a similar manner between two stages (+5-6 dB

and 2-3 dB). No KAZR observations were conducted at Nauru.

The ARM NSA and SGP sites are the two longest operating sites of the ARM program ([Sisterson et al., 2016]; [Verlinde et al., 2016]). The NSA represents a typical Arctic environment with very low temperatures while the SGP has been the observational centerpiece and anchor of the ARM Program since 1992. The calibration offsets for the period 2008 to 2017 at

these two sites along with the ARM intramode differences are shown in Fig. 10. The NSA MMCR 2 significantly overestimates the radar reflectivity and a calibration offset between -4.4 to -8.4 dB (gradually increasing from 2008 to 2009) is required to minimize the RMSE when compared to CloudSat. This large calibration offset is consistent with the impact of corrosion on



the waveguide that was attached to the antenna feed effectively breaking the connection between the waveguide and the feed. This hardware failure went unobserved until during one of the visits of the ARM program radar engineer was accidently discovered during a system inspection ([Kollias et al., 2016]). During the same period, the ARM intramode difference (Mode 3 – Mode 2) gradually increases from 0.8 to 2.5 dB. The NSA KAZR MD mode is compared to Cloudsat for the period 2012

to 2017. During the first 2 years, the KAZR MD calibration offset is for the most part within ± 1 dB suggesting that the radar was well calibrated. During the 2014-2017 period, the KAZR MD mode calibration offset is between +3 to +6 dB and the ARM intramode (GE – MD) difference is around -1.7 dB. The SGP MMCR mode 2 calibration offset is significant during the period 2008-2011. In 2008 the calibration offset is between +7 and +10 dB, -3.5 to -4.5 dB in the early part of 2009 and +4 to +6 dB for remaining of the operating period of the MMCR at the SGP. The ARM intramode difference (Mode 2 – Mode 3) is

for the most part between +0.5 to 0.9 dB. The SGP KAZR MD mode is compared to Cloudsat for the period 06/2011 to 12/2017. The calibration offset values are positive (+3 to +6 dB) at the beginning and then negative (-1 to -6 dB) during the 2014-2017 period. The ARM intramode differences (GE – MD) are between ± 1 dB and small shifts in their magnitude and sign correlate with periods where the calibration offset changes.

The ARM Mobile Facility (AMF) is a portable atmospheric laboratory equipped with a sophisticated suite of instruments designed to collect essential data from cloudy and clear atmospheres in important but under-sampled climatic regions. As such, the AMF deployments are often the only source for ground-based observations of clouds and precipitation at some of the AMF deployments ([Miller et al., 2016]). Here, we report the calibration offsets for 5 deployments of the first ARM Mobile Facility (AMF1) and 2 deployments of the second ARM Mobile Facility (AMF2). The results are shown in Fig. 11. The AMF1

deployments are: Niger, West Africa (NIM), Black Forrest, Germany (FKB), Graciosa island, Azores (GRW), Cape Cod, Massachusetts (PVC), and Manicaparu, Brazil (MAO) and the AMF2 deployments are: Hyytiälä, Finland (TMP) and McMurdo Station, Antartica (AWR). The AMF deployments are typically one-year deployments, except for the GRW and MAO deployments that lasted for two years. At the AMF1 deployments the main profiling cloud radar system was the WACR and at the AMF2 deployments a KAZR. The short duration of the mobile deployments coupled with the time needed to relocate

the AMF's to their next field location makes the AMF calibration offset record sparse. At NIM, the AMF deployment was over 13 months long but the WACR was deployed for only 8 months and two WACR calibration offset are estimated (+4.4 and +4.0 dB). The following year, during the FKB AMF deployment, four WACR calibration offsets are estimated (+3.7, +3.8, +2.8 and 2.4 dB). During the 2-year deployment at GRW the WACR calibration offset started from a low value of +1.4 dB and gradually rise to +3.2 dB. At PVC, the WACR calibration offset was between +3.3 to +3.5 dB. Noticeably, the WACR

calibration did not change a lot after a number of field deployments including two in Asia (China and India) were not sufficient record of WACR observations are available to conduct an ARM – CloudSat comparison. However, during the 2-year MAO deployment the estimated calibration offsets are higher and more variable (+3.9 to +8.5 dB).

The AMF2 was established later than the AMF1, thus, its deployment record is shorted. The AMF2 deployment Hyytiälä,

Finland (TMP) has been considered as the first successful deployment of triple frequency radar observations by the ARM program ([Kneifel et al., 2015]) with well calibrated radar systems. The ARM – CloudSat comparison confirms that the KAZR MD mode was well calibrated during the TMP deployment and the calibration offsets are -0.2, +1.0 and +1.6 dB (Fig. 11).



During the most recent AMF2 deployment at McMurdo Station (AWR), significant calibration offset is found. Due to surrounding elevated topography, AWR is the only site where additional post-processing of the CloudSat observations was required to eliminate antenna sidelobe contributions. In addition, the AWR high latitude location in combination with the restriction of daylight only CloudSat observations limited the number of available CloudSat samples especially during the southern hemisphere winter (Fig. 5). As a result, most of the CloudSat samples are available at the beginning and the end of the field campaign. At the beginning, the calibration offset is +7.7 dB and during the latter part of the mobile deployment is between +3.5 and +5.1 dB (Fig. 11). The ARM intramode difference (GE – MD) is -1.2 dB at the beginning of the period and -0.65 dB later in the deployment time period.

## 4 Summary

The DOE ARM program has been at the forefront of the development and application of profiling and scanning millimeter wavelength radars for over 20 years. The long record of ARM cloud radar observations represents a unique dataset that provides a bottom-up, high resolution view of clouds and precipitation at a number of locations around the globe. Calibration is an important step in utilizing the full potential of these observations for the evaluation of numerical models and the development of quantitative microphysical retrievals. The use of CloudSat as a global calibrator for cloud radars was first proposed by [Protat et al., 2011]. Here, the [Protat et al., 2011] technique is revised, improved and automated and the entire record of CloudSat observations (2007 – 2017) is used to provide a calibration reference for over 43 years of ARM profiling cloud radar observations at fixed and mobile sites. A total of 653 ARM – CloudSat comparisons are performed using a running 6-month time window.

The application of the ARM – CloudSat comparison methodology across the entire network of ARM profiling cloud radars is not straightforward. Four generations of ARM cloud radar systems, operating at two different radar frequencies (35- and 94-GHz) are evaluated. All the radar systems (with the exception of the AMF1 WACR) operate using a sequence of modes with difference capabilities in order to provide a uniform radar sensitivity and performance throughout the troposphere. The offset in the reported radar reflectivity by these different modes for each radar are documented as a function of time. Abrupt changes in the offset magnitude and sign are found to correlate well with changes in the radar calibration as deduced by the statistical comparison with CloudSat. Furthermore, the geographical location, the seasonal variability of the clouds and precipitation occurrence and the operational status of the CloudSat CPR significantly affect the number of samples available within a 6-month time window to perform the ARM – CloudSat comparison. When the number of CloudSat columns is less than 500-1000 the comparison is difficult to perform. Out of the possible 653 calibration coefficients, 616 were accepted, a 94.3% success rate.

The analysis demonstrates that both historic (i.e., MMCR) and recent ARM radar operations (i.e. KAZR2) require considerable adjustments before they can be used in a quantitative way. The analysis from [Protat et al., 2011] and the experience gained in



this study using the technique in a much largest dataset suggest that accuracy of the CloudSat-based calibration of ground-based cloud radar systems is accurate within 1-2 dB. In most cases, the observed calibration offsets exceeded this uncertainty value suggesting the estimated calibration coefficients should be considered in a future reprocessing of the ARM radar record. Furthermore, the gradual temporal change in the observed calibration offsets, the correlation of large swings in the calibration offset with periods when the ARM radar hardware and/or software was not operating in an optimum way, suggest that the use of CloudSat can provide reliable information that can be used to characterize the calibration of ground-based radar systems. Planned and future spaceborne radar systems such as the Earth Clouds Aerosols and Radiation Explorer (EarthCARE, [Illingworth et al., 2015]; [Kollias et al., 2018]) or future spaceborne radar concepts (Tanelli et al., 2018) will provide similar spaceborne radar measurements to evaluate large profiling cloud radar networks (e.g., ARM, ACTRIS) in the future.

## 5 Acknowledgements

P. Kollias and B. Puigdomènech Treserras were supported by the US DOE ARM and ASR radar science project. The authors would like to thank Jim Mather, Bradley Isom and Joseph Hardin for reviewing the manuscript and providing valuable feedback. All ARM data streams are available online at: http://www.archive.arm.gov/discovery/.



## Appendix A

The use of surface-based measurements of the raindrop PSD using impact or optical disdrometers to calibrate profiling and scanning precipitation radars is not new. This technique has been widely used in the past for calibrating profiling cm-
wavelength Doppler radar ([Gage et al., 2000]; [Tridon et al., 2013]). In the case of cm-wavelength radars, wet radome or antenna attenuation is negligible, the systems are configured to have sufficient dynamic range to detect intense precipitation returns without receiver saturation and the Rayleigh scattering approximation is valid in most cases. At mm-wavelength radars, several factors need to be considered: the wet radome can induce considerable attenuation, at high rain rates the Rayleigh scattering approximation is not valid and receiver saturation occurs at lower rain rates. Here, we use the Parsivel2 disdrometer
(OTT Hydromet GmbH) measurements. The disdrometer provides 1-min averaged raindrop PSD's. From the Parsivel2 files, the variable "equivalent_radar_reflectivity" which is the radar reflectivity calculated by the ARM ingest is used. All 1-min Parsivel measurements where raindrops with diameter > 4.5 mm are detected are filtered out to avoid the impact of false detection of large raindrops in the Parsivel2 - KAZR2 comparison. The Parsivel2 time assigned to each data point indicates the beginning of a 1-min period of averaging. Using this time, 1-min averages of the KAZR2 reflectivities in linear units are
estimated. Next, the KAZR2 radar reflectivities are corrected for path attenuation induced by the hydrometeor. The relationship $A\,(\mathrm{dBkm^{-1}}) = 0.28 \cdot R\,(\mathrm{mmhr^{-1}})$ is used to estimate the one-way attenuation at Ka-band ([Matrosov, 2005]). Only the 1-min data when the Parsivel2 radar reflectivity is between 0 and 20 dBZ are used. The lower limit is used to ensure that the Parsivel2 samples enough raindrops. The upper limit is used to minimize the impact of wet radome attenuation and to ensure that the Parsivel2 radar reflectivity estimates using the Rayleigh scattering approximation have no or negligible non-Rayleigh
effects. The KAZR2 and Parsivel2 radar reflectivity time series were investigated for possible time lag, however, given the proximity of the radar data to the ground, no significant time lag was found. Finally using a running time window of 90 days, the mean of the differences of the KAZR2 and Parsivel2 radar reflectivities is estimated.

Fig. A1a shows the time series of the calibration offset between the Parsivel2 and the KAZR2 for different KAZR2 range
gates. In general, the calibration offset is positive, thus, implies that the KAZR2 underestimates the radar reflectivity. However, the calibration offset varies a lot with the range gate. The KAZR2 is a pulsed radar, thus after each pulse transmission the receiver protection circuit (T/R switch network) needs to switch from transmit (closed receiver) to receive (open receiver) mode. The switch takes several hundreds of nanoseconds, thus, the KAZR2 returns from the first range gates (3 to 7) report lower radar reflectivity values, resulting to higher radar calibration offset values. Our analysis identified range gate 8 (240 m)
as the closest range gate to the surface that is unaffected by the KAZR2 T/R switch network. Above range gate 8, the calibration offset continues to decrease, highlighting the impact of the evaporation in modifying the raindrop PSD. The scatter plots between the KAZR2 radar reflectivity at range gate 8 and the Parsivel2 radar reflectivities during the two extensive periods are shown in Fig. A1b, c. These two periods match the periods used to estimate calibration offsets using the ARM – CloudSat comparison technique (Fig. 8). The ARM – CloudSat comparison indicated calibration offsets of 0.3 and 5.2 dB and the ARM
– Parsivel2 comparison indicated calibration offsets of 0.57 and 3.91 dB.

Disdrometers have certainly the potential to monitor the calibration of profiling cloud radars and this topic warrants additional analysis using comprehensive datasets from different cloud radar systems and for different climatological conditions. For example, Frequency Modulated Continuous Wave (FMCW) radars ([Küchler et al., 2017]) do not have T/R switch networks,
but careful analysis is required to ensure proper alignment of the two antennas or correct for the antenna parallax problem





([Sekelsky and Clothiaux, 2002]). Furthermore, careful analysis is required to avoid using radar returns that saturate the radar receiver especially at short ranges and to account for non-Rayleigh scattering in case of 94-GHz radar systems. This careful analysis is beyond the scope of this study.





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



**Table 1: Information on the sites (name and location), the type of radar and the start and end date of the calibration period**

| Site | Radar | Lat (°); Long (°) | Start time [mm/yyyy] | End time [mm/yyyy] |
|---|---|---|---|---|
| **ENA** | KAZR2 | 39.09;  -28.03 | 12/2015 | 12/2017 |
| **OLI** | KAZR2 | 70.49;  -149.88 | 07/2015 | 12/2017 |
| **AWR** | KAZR | -77.85;  166.73 | 12/2015 | 01/2017 |
| **NSA** | KAZR | 71.32;  -156.62 | 10/2011 | 12/2017 |
| **NSA** | MMCR | 71.32;  -156.62 | 01/2008 | 12/2009 |
| **SGP** | KAZR | 36.60;  -97.48 | 02/2011 | 12/2017 |
| **SGP** | MMCR | 36.60;  -97.48 | 01/2008 | 12/2010 |
| **TMP** | KAZR | 61.84;  24.29 | 01/2014 | 09/2014 |
| **MAO** | WACR | -3.21;  -60.60 | 03/2014 | 12/2015 |
| **PVC** | WACR | 42.03;  -70.05 | 10/2012 | 06/2013 |
| **GRW** | WACR | 39.09;  -28.03 | 01/2009 | 12/2010 |
| **FKB** | WACR | 48.54;  8.40 | 04/2007 | 12/2007 |
| **NIM** | WACR | 13.48;  2.18 | 04/2006 | 12/2006 |
| **TWP-Darwin** | KAZR | -12.42;  130.89 | 02/2011 | 04/2014 |
| **TWP-Darwin** | MMCR | -12.42;  130.89 | 01/2007 | 02/2011 |
| **TWP-Manus** | KAZR | -2.06;  147.42 | 10/2011 | 10/2013 |
| **TWP-Manus** | MMCR | -2.06;  147.42 | 05/2007 | 03/2011 |
| **TWP-Nauru** | MMCR | -0.52;  166.92 | 05/2007 | 02/2009 |





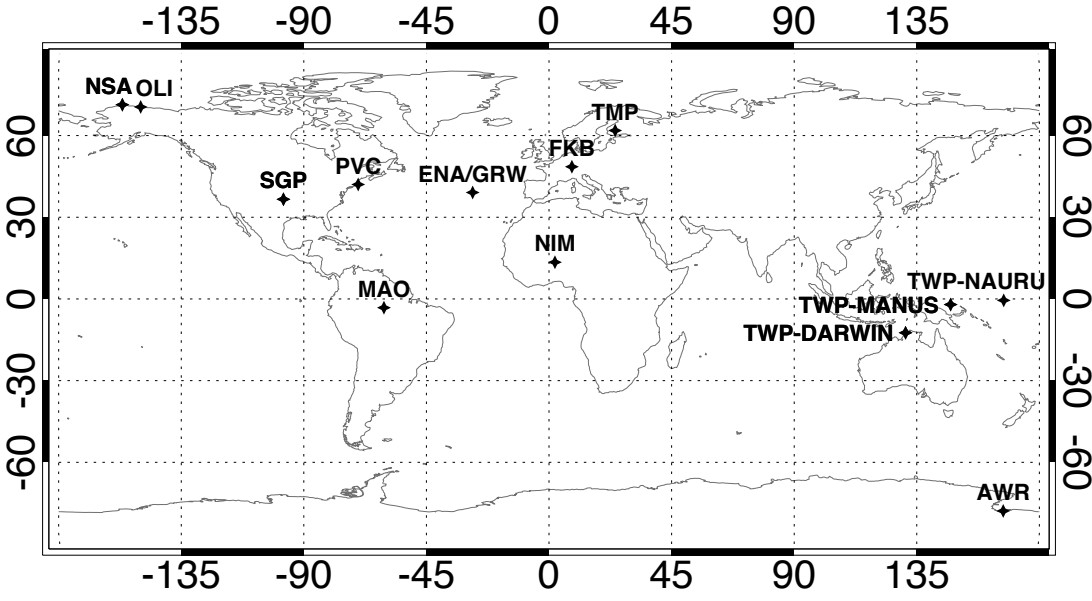

**Figure 1: Location of the fixed and mobile ARM profiling cloud radars calibrated using the CloudSat Cloud Profiling Radar**





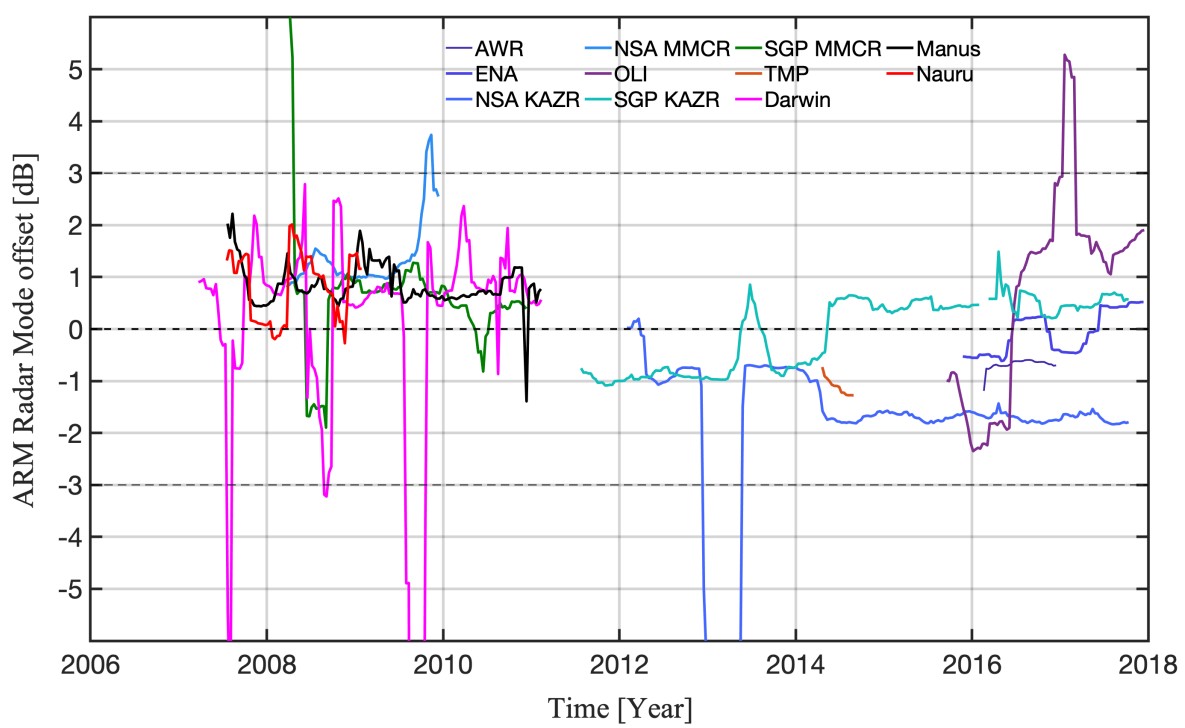

**Figure 2: The difference [dB] in the radar reflectivity reported between different ARM modes. For KAZR/KAZR2 systems the GE – MD difference and for MMCR systems the Mode 3 – Mode 2 differences are reported.**





Figure 3: Algorithm flowchart of the calibration of the ARM Cloud Radars using CloudSat Observations




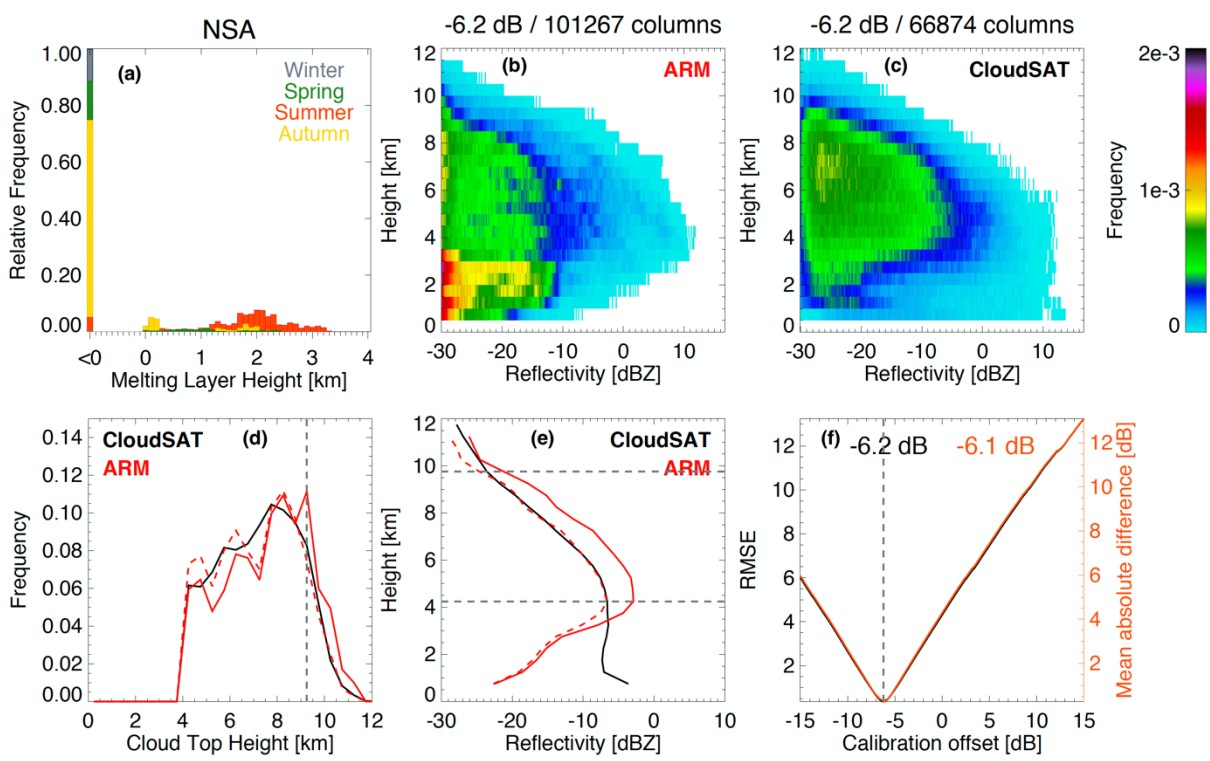

**Figure 4: Example of the CloudSat – ARM radar calibration at the NSA for mode 2 of the MMCR for the period 01/2008 to 12/2009.**
**(a) the seasonal distribution of the melting layer heights, (b) the distribution with height of the ARM radar reflectivities, (c) the**
**distribution with height of the CloudSat radar reflectivities, (d) the comparison of the CloudSat (black) and ARM (red) cloud top**
**height histograms. The solid red line indicates the cloud top height distribution from the archived ARM radar data and the dashed**
**red line indicates the cloud top height distribution after 6.2 dB is subtracted by the ARM radar reflectivities, (e) the comparison of**
**the CloudSat (black) and ARM (red) mean radar reflectivity profiles. The solid red line indicates the ARM mean radar reflectivity**
**profile using the archived ARM radar data and the dashed red line indicates the ARM radar reflectivity profile after 6.2 dB is**
**subtracted by the ARM radar reflectivities, (f) the RSME value between the radar reflectivity profiles for different calibration**
**offsets.**





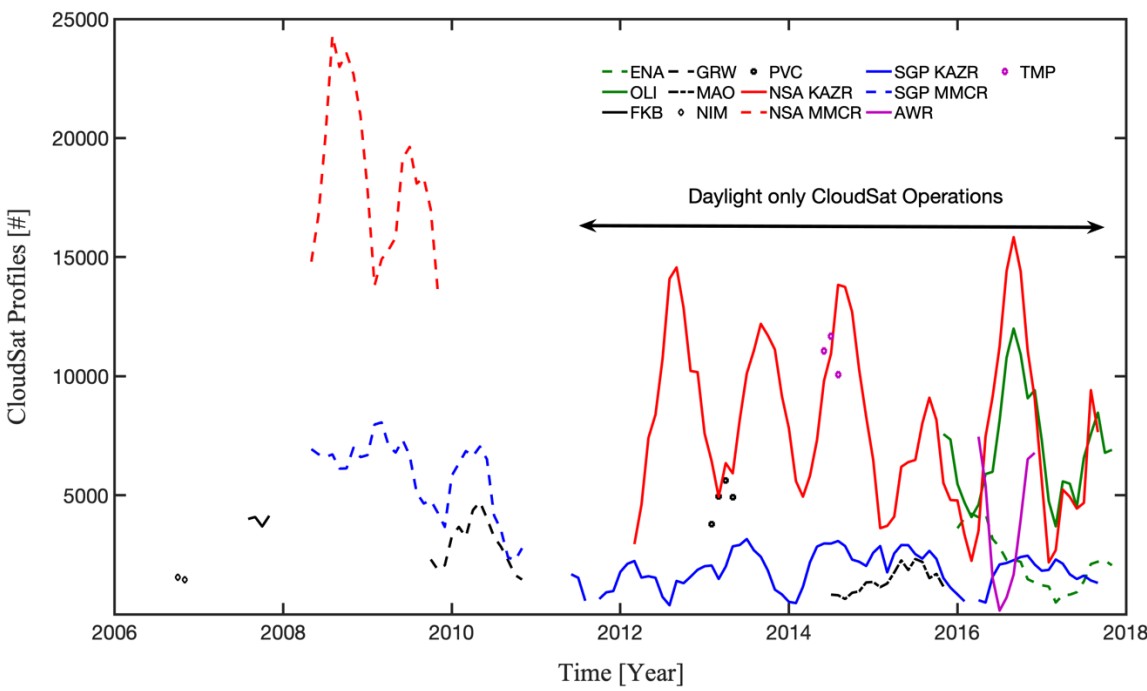

**Figure 5: The number of CloudSat profiles found within a ± 3 month window to be suitable for calibrating the ARM cloud radars as a function of time for the different ARM sites.**

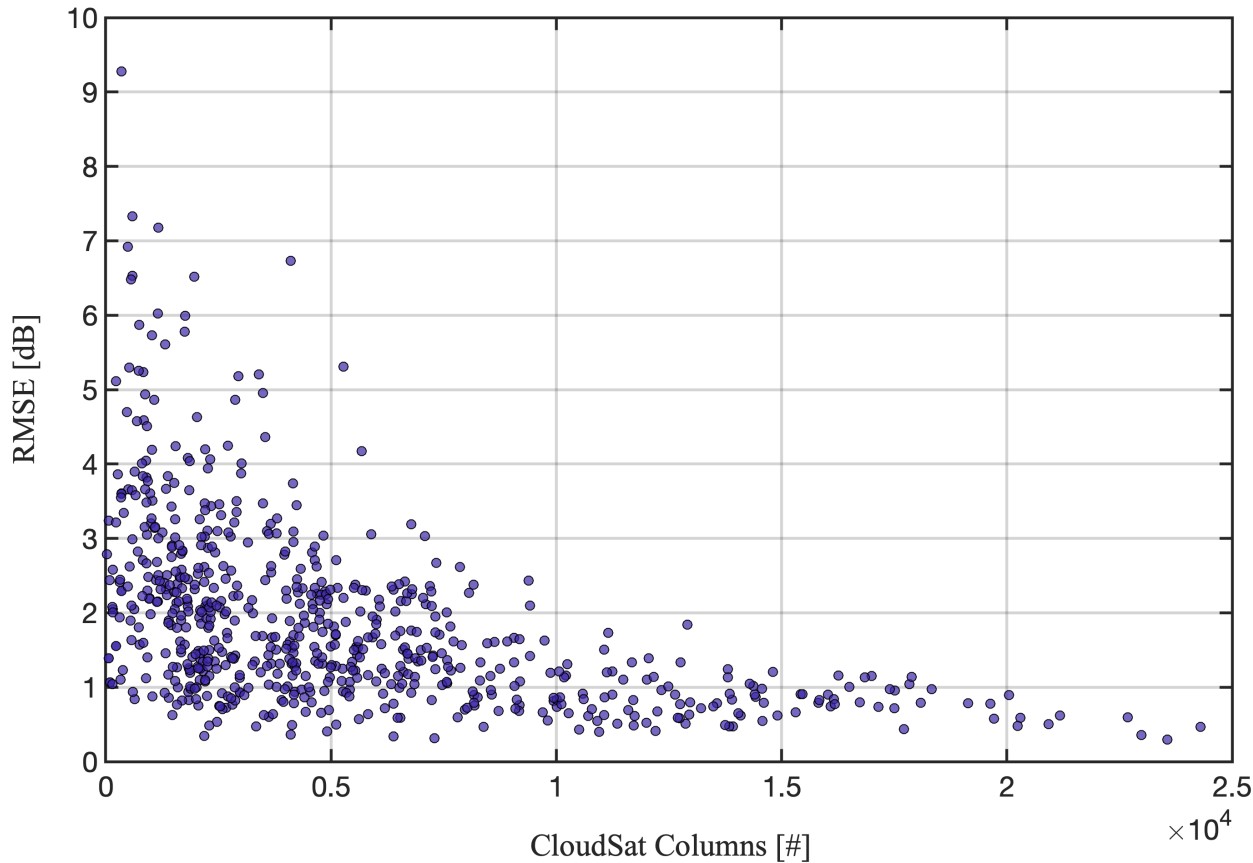

**Figure 6: The relationship between the minimum RMSE [dB] achieved in a particular ARM – CloudSat comparison and the**
15 **corresponding number of CloudSat columns.**





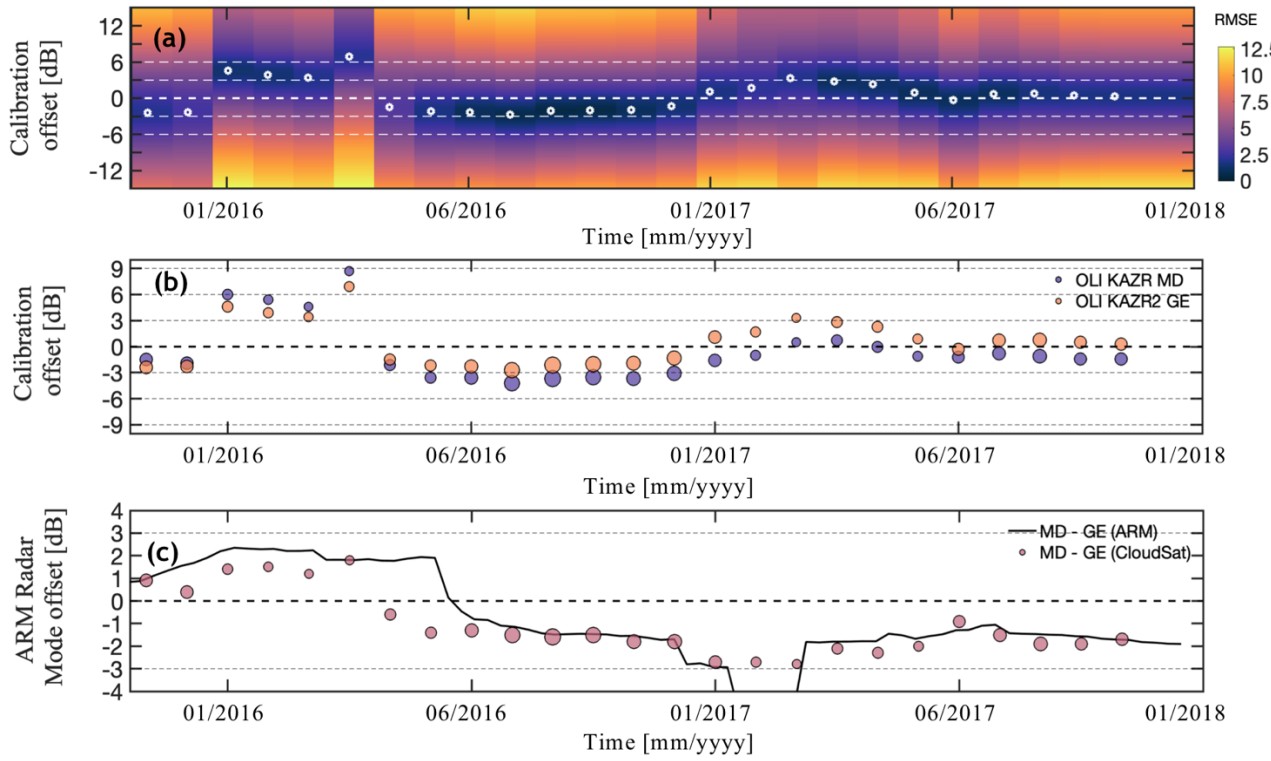

**Figure 7: (a) The OLI KAZR2 MD mode calibration offset as reported by the CloudSat-ARM comparison (circles). The colors indicate the RSME of the CloudSat-ARM comparison for different radar calibration offsets, (b) The AMF3 (Oliktok Point, Alaska) KAZR2 MD and GE modes calibration offset as reported by the CloudSat-ARM comparison. The size of the circles indicates the ratio of the sample size of the CloudSat columns for any given calibration offset estimate relative to the maximum sample size of**
15 **CloudSat columns observed during the same period by the same mode, (c) The difference between the MD-GE modes as reported by the CloudSat-ARM comparison and as reported by the ARM radar mode to ARM radar mode comparison.**





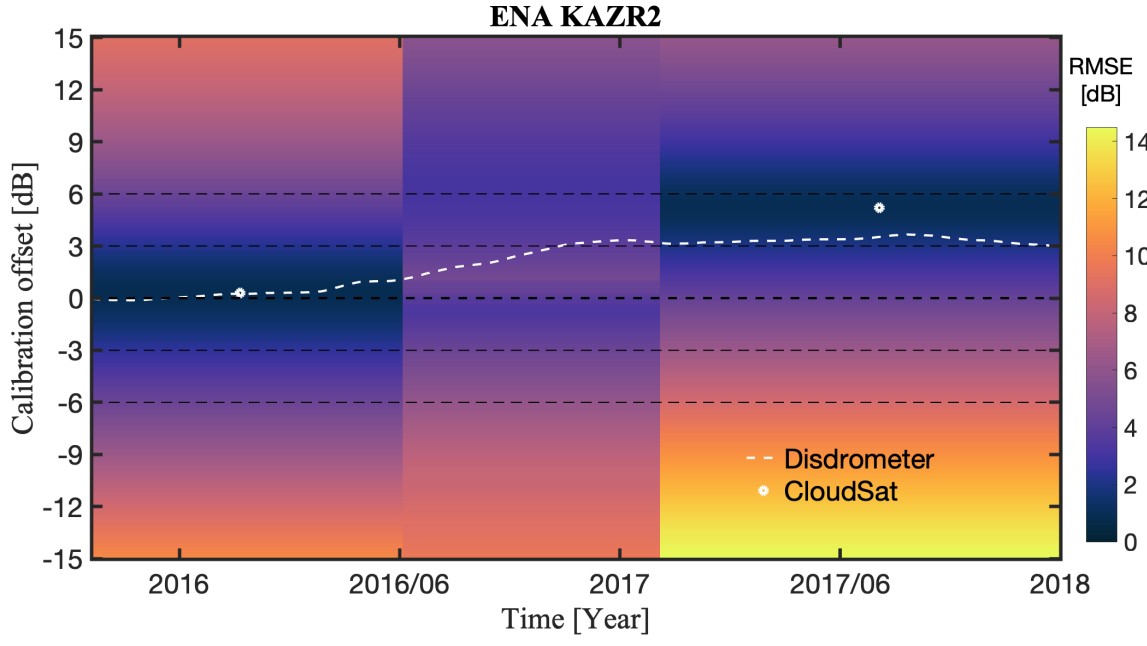

**Figure 8:** **The ENA KAZR2 calibration offset as reported by the CloudSat-ARM comparison (star symbol) and by the KAZR2 and Parsivel disdrometer comparison (line). The colors indicate the RSME of the CloudSat-ARM comparison for different radar calibration offsets.**





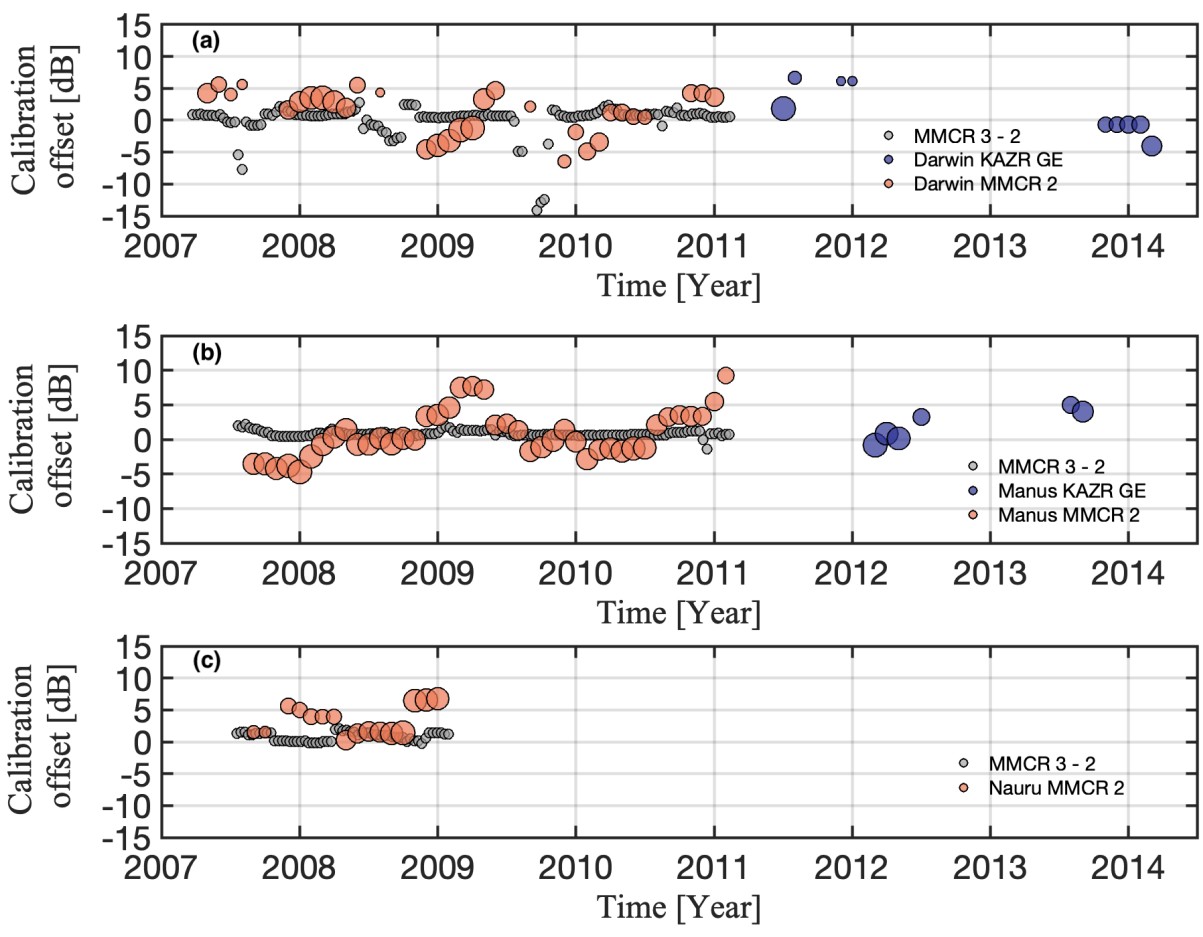

**Figure 9: The calibration offset for the MMCR mode 2 and KAZR GE mode at the Tropical Western Pacific (TWP) sites of (a)**
10 **Darwin, (b) Manus and (c) Nauru based on the ARM – CloudSat comparison. The size of the circles indicates the ratio of the sample size of the CloudSat columns for any given calibration offset estimate relative to the maximum sample size of CloudSat columns observed during the same period by the same mode. The gray circles indicate the ARM mode 3 -mode 2 difference as estimated from the ARM radar mode intercomparison.**

**Figure 10: The calibration offset for the MMCR mode 2 and KAZR GE mode at (a) NSA and (b) SGP sites based on the ARM – CloudSat comparison. The size of the circles indicates the ratio of the sample size of the CloudSat columns for any given calibration offset estimate relative to the maximum sample size of CloudSat columns observed during the same period by the same mode. The gray circles indicate the ARM MMCR mode 3 -mode 2 and KAZR GE -MD mode difference as estimated from the ARM radar mode intercomparison.**



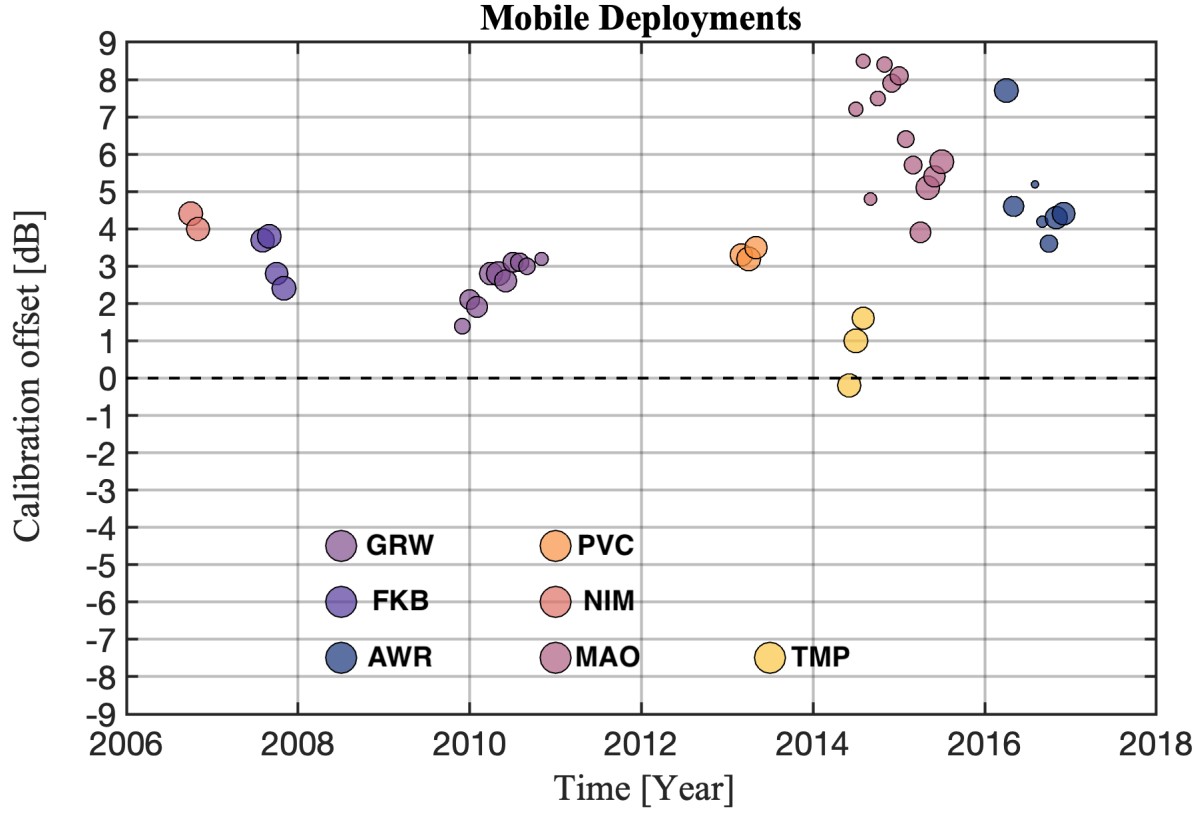

**Figure 11: The radar calibration offset we should add to the reported ARM cloud radar reflectivities in order to minimize their**
10  **differences with those reported by the CloudSat CPR at the ARM Mobile Facilities (AMF) sites. The size of the circles indicates the**
**ratio of the sample size of the CloudSat columns for any given calibration offset estimate relative to the maximum sample size of**
**CloudSat columns observed during the same period by the same mode.**





**Figure A1: (a) The calibration offset between the KAZR2 and Parsivel2 estimated using KAZR2 measurements from difference range gates (from the 3rd to the 20th), (b) the calibration offset for the period 01/01/2016 to 06/01/2016 using the 8th KAZR2 range gate and (c) the calibration offset for the period 01/01/2017 to 10/01/2017 using the 8th KAZR2 range gate. These two periods**
5   **correspond to the periods used for the ARM - CloudSat calibration offsets shown in Fig. 8.**