# Peer review of "Calibration of the 2007-2017 record of ARM Cloud Radar Observations using CloudSat"

_Atmospheric Measurement Techniques, 2019_

## Referee Comment (RC1) · Anonymous Referee #1 · 10 Apr 2019

Review of the article titled "Calibration of the 2007-2017 record of ARM cloud radar observations using CloudSat" by Kollias and coauthors for publication in AMT. The authors have compared the reflectivity from vertically pointing Doppler cloud radars at the ARM sites to the reflectivity from radar onboard polar orbiting satellite. The goal of this study is to characterize the performance of the ground-based radars, as the space-borne radar is well-calibrated. They find significant calibration offset for the ground-based radars throughout the 10-year period, and inherent inconsistencies between the different modes of them. The technique is already well-established and used here in a relatively straightforward manner. The article is overall okay, but needs several small tweaks in writing. Due the number of small corrections listed below, I recommend this

[Figure]

article for major revisions.

Major Suggestions:

1) As the authors have already made CFAD of all the ARM radars, it will be relatively straightforward for them to calculate the minimum detectable signal (MDS) for them. You can just pick up the bottom 5% of reflectivity at 1 km, and make its average. This will greatly assist the scientific user community, as it is unclear how sensitive are the ARM radars and if their sensitivity has changed through years. You already have the data for calculating this and hence will be a worthy and useful effort. If you do this, then you can add this as another column in Table 1. Thanks.

2) Please add one more column to Table 1 and report the average and standard deviation of the calibration offset for each radar. This table will be very useful for users who'd like to use your calibration offsets in their research. Please add the different modes of KAZR and MMCR in the rows. I understand that this will be an average through many years, but still worthy of reporting. Thanks.

3) The article ends abruptly and you only provide a brief summary without discussing the implications of your results. It will be good if you can devote one paragraph each on the following two things i) the impact of calibration offset on the regular data products produced by the ARM program. I did a quick search and the radar reflectivity is used for doing microphysical retrievals like MicroBase and cloud drop concentrations. Please discuss how a calibration offset might affect these data-products. ii) The lead author has significant expertise in retrieving vertical air motion and microphysical properties from ground-based radars. A quick search made me realize that scientists have also used radar reflectivity in those studies in addition to using mean Doppler. It will be good if you can elaborate on how your results will impact the results previous studies by you and from Giangrande, Verlinde, Luke, Shupe, Dong, Chiu etc. So please add two separate paragraphs at the end and rename the section as "Summary and Discussion".

Minor Issues:

It is unclear to me if the authors are referring to the funding agency ARM or their observatories through the user facility. I recommend using the ARM Climate User Facility throughout the article. Thanks.

Page-1, Line 16: Add "collectively" before "Over". Thanks.

Page-1, Line 12: Add :1990s

Page-1, Line 15: the sentence doesn't read well, please rephrase.

Page 2, Line 8: "Surprise" not "surprised".

Page 3: Add outline of the paper before the section 2.

Page 3, Line 19: "At couple of sites".

Page 4, line 1-2: Please rephrase and remove "us".

Page 4, Line 15: "Computed" and not "computer".

Page 5 Line 10, Page 7 line 13: It is unclear which numbers to believe.

Page 5-6: It will be good if you mention the equation used for doing gaseous correction in CloudSat. Thanks.

Page 8 Line 15: "were" not "where".

Page 10 Line 15: you mean "observatory" and not "laboratory"?

Page 11 Line 20: there is a typo, it should be 616 samples according to Page 7 line 32. Thanks.

---

## Referee Comment (RC2) · Anonymous Referee #2 · 2 May 2019

The present manuscript describes the efforts of the Authors to calibrate a long series of ground-based radar measurements using space-borne radar measurements from CloudSat. This task is all the more important as it can affect the quality of atmospheric retrievals. Moreover, the calibration of such a long time series on a common ground helps greatly the study of the climate on such time scales The article provides in-depth information into the operation and maintenance of the ARM radar network. As such, it makes publicly available information that otherwise would be known only to the few expert users/members of the ARM program. For that alone, this manuscript is worth publishing. The Authors follow a clear path to describe their datasets, its quality control, and the methods to collocate and optimize the calibration assessment. Various

graphs provide a nice illustration of the performance of the proposed method. Before publishing this article, these are the points I would like the Authors to address: 1. The article needs a serious editorial revision to correct for grammar errors and typos. In particular, the Authors should revise the tenses of the verbs for consistency. 2. Please provide a table listing the various acronyms, and please define these acronyms in the article at their first occurrence. 3. As a general question, would the statistical method that you use (to match the mean profiles) work if you also match the envelope of the CFAD (lower and upper quantiles)? This envelope may have useful information, e.g. on the variability of the reflectivity profile over time or space... 4. Would the Authors see any merit/advantage in applying their optimal calibration method to other satellite datasets collocated with ARM radars? Could you please comment on this in you article?

Please also note the supplement to this comment:
https://www.atmos-meas-tech-discuss.net/amt-2019-34/amt-2019-34-RC2-supplement.pdf

---

## Referee Comment (RC3) · Roger Marchand (Referee) · 5 May 2019

Review of Manuscript amt-2019-34

Title: Calibration of the 2007-2017 record of ARM Cloud Radar Observations using CloudSat

Authors: Kollias, Treserras and Protat

Overview:

I think publication of this technique and the results for ARM radars will be of value to many investigators and investigations that have (and will continue to) rely on ground-

based radar datasets. While I was under no illusion that the ARM radars were well calibrated, I nonetheless found the results to be sobering.

Recommendation: Publish after minor revisions

General Comments:

1) A few more details on the technique.

I largely follow the technique, but you need to add a few more details, see specific comments for Page 6. The goal should be to make it so that someone else could implement the approach given this description. In particular, please discuss uncertainties in estimated calibration corrections associated with equations 2 and 3, as well as the height range used to estimate the best offset.

2) Differences in rules and thresholds for including or not including columns & Verification metrics.

Do the different rules and thresholds for including or not including radar columns, which are to some degree necessarily different between CloudSat and ARM, matter? (see e.g. differences on Page 5, line 16; Page 7 line 17). I am concerned about the possibility that differences in the mean Z-profiles might be due simply to having different conditions or "distributions of cloud-types" in each collection from which the mean-Z-profile is calculated.

One way to check this would be to look not just at the mean-Z-profile but also to ask if the two profiles are based on a similar fraction of the observations in each set. Said another way, once you construct your "non-precipitating CFAD" and pick your dBZe-threshold (for calculating the mean-Z-profile), is the profile of cloud fraction associated with this dBZ-threshold the same for both CloudSat and ARM. If it is, then one can be confident that errors in the reflectivity correction due to differences in cloud populations will be small.

I suggest creating a metric, such as the vertically integrated absolute cloud-fraction

difference divided by the mean cloud fraction, and plotting this information along with the calibration corrections. Likewise, it would be interesting to see how this metric depends on the number of columns, which like figure #6, should give one a sense of what is a reasonable value for this quantity.

Likewise, the cloud-top-height (CTH) distribution comparison you introduce (Fig 4d) provides confidence that the calibration correction is robust and that it is based on the same cloud populations. As far as I can see, after you introduce the idea of this as "a verification", you don't use it. At a minimum it seems like you should discuss whether the CTH distributions are consistently improved (made more similar) with the radar correction or not. Again, you might make a metric that expresses this improvement – though I suspect the above cloud fraction metric is likely better for this purpose.

3) Results for Darwin and the size of the analysis region

I don't typically like to point to my own work when reviewing an article, but in this case I think some work that a former student of mine Zheng Liu, Tom Ackerman and I have done at Darwin is very germane to this study.

Liu, Z., R. Marchand, and T. Ackerman (2010), A comparison of observations in the tropical western Pacific from ground‐based and satellite millimeter‐wavelength cloud radars, J. Geophys. Res., 115, D24206, doi:10.1029/2009JD013575.

In particular, we compared CloudSat and the Darwin MMCR measurements and we investigated the size of the analysis region and sampling uncertainties in some detail. That study very much supports using a 300 km radius area and 6 month window.

Note also figure 8 in this paper. While we did not derived a calibration offset, our results are broadly consistent with idea that ARM calibration was too LOW at Darwin in the 2006-7 (wet season), and agrees your with figure 9a (CloudSat – ARM difference of ∼= 5 dB at this time).

Minor Comments:

Page 1, line 28. detail => detailed

Page 2, line 5. Opening sentence is awkward, rephrase. Perhaps "Part of the motivation for the ARM radar expansion, was to improve cloud microphysical retrievals through the use of dual-wavelength ratios, that is, making use of the relative difference in radar scattering at different wavelengths. This difference signal is often only a few dB and as one might expect, this .... "

Page 2, line 11. Do you mean calibrating ARM vertically pointing radars is more difficult than the WSR-88D network? What is being compared to what is not clear? Suggest you rephrase this and following sentence to be clearer and generally read better.

Page 3, line 3. Change "... is such diverse" to "... to such a diverse set ...".

Page 3, line 31. "...on the same..." to "at the same".

** Page 4: First paragraph: Mode analysis.

How did you account for differences in the minimum detectable signal between the modes? I presume you only included neighboring time & range bins where both modes have a measurement with high SNR?

Page 4. The first paragraph launches into a discussion of mode differences (which is useful) but a bit confusing when one is expecting a comparison of CloudSat and ARM calibrations. I suggest breaking this paragraph about line 8 and adding.

"Therefore as a prelude to comparing CloudSat and ARM, we begin with a comparison of reflectivity values between ARM radar modes. As will become clear later, changes in the reflectivities between modes is often, though not always, indicative of changes in overall calibration."

Page 4, line 13. I presume "bid" should be "big". Perhaps

Page 4, line 14. Specify period (6 months?).

Page 4, line 16. Perhaps change to read: "Overall, the mode reflectivity differences are small (±2 dB) and only occasionally are the differences much higher than 2 dB. While the absolute values of mode difference is in the next generation of ARM cloud profiling radars (KAZR and KAZR2) is often similar, arguably there are fewer jumps or rapid changes (except perhaps at OLI). In general it is difficult to identify which mode has a better calibration, because as will be shown, the calibration difference between CloudSat and ARM is typically larger than ±2 dB.

** Page 4: Second paragraph on difference between Protat and current approach.

This paragraph is nearly impossible to follow if you don't already know what Protat 2010 did. In particular, I have no idea what "... a rigorous selection of the CloudSat overpasses within a certain radius to avoid any errors in the estimation of the proximity of CloudSat columns to the ARM site location" means. But other parts of this are confusing to me (and I am familiar with Protat 2010). I strongly encourage you to reorganize the manuscript such that you FIRST explain your approach in detail and ONLY at the end of this material highlight how this approach differs from Protat (2010).

** Page 6, equation 2. What is the justification for using a constant here? You used Rosenkranz and ...? Somehow you must have specified some set of atmospheric profiles to come up with these constants? Explain in enough detail so someone else could implement this idea. Nominally, I think it would have been better to calculate a set of gasses corrections for ARM (perhaps using ERA data just as CloudSat does). **But in lieu of this, I think you need to address how much error (uncertainty) using this constant introduces in your calibration correction.

** Page 6, equation 3. What assumptions does this equation entail? Again, I think you need to address how this impacts the uncertainty of your calibration corrections?

** Page 6, line 17. I am not sure I understand this definition, there seems to be a grammatical error here. Do you simply mean "precipitaing column = 10% (or more) of the radar volumes below the FL have ANY reflectivity (even if it is -30 dBZe). So if FL

is 2 km at there are 20 bins below 2km, if 3 bins have ANY detection (even if it is just a low cloud) you are calling it precipitating?

** What do you do if the FL is near the surface (in the CloudSat clutter) or with only a few ARM bins $\sim$ 5 ?

** Why do you use a different threshold 35% for CloudSat? This seems arbitrary.

** Page 6, items 2 to 4. This material seems important and needs to be better explained. In particular, how does the "degradation" work? I presume you mean that the mean-Z-profiles, are obtained from the CFADS in step 4 by summing bins with dBZ > Threshold > Minimum Detectable Signal (MDS) weighted by the bin dBZe? (I note without weighting this just give you the "profile of cloud fraction"). If yes, it might be important to choose threshold that is + 3 to 5 dB larger than the MDS.

** Where/How does the SNR > -15 mentioned early come in?

**Page 7, item 2. My experience at Darwin suggests (and your example in Fig. 4) that the height range used might matter here. How much does the estimated correction change in this example if you change the range form 3 to 12 km, that is, 10 to 12 Km and 3 to 4 km?

Page 7, line 13. What does "maximum fraction of ... warm temp" mean ? I don't follow.

Page 7. It seems you address the issues of the number of columns in detail later in the text, but do NOT the distance issue. (see also general comment #3). Perhaps add some discussion and/or better yet show result for OLI site (where you have lots of data) – add a line to fig 7 – for results based on 100 vs. 300 km?

** Page 7/8, analysis on number of columns vs. number of good columns ?

I like very much the analysis you have included on the number of columns. But unless I misunderstand you are counting ALL columns here. Not the number of good columns (i.e. columns which are devoid of high/ice clouds or precipitating). I think it would be

far more sensible to count the number of columns with good data (and set a minimum threshold on this) rather than all radar columns.

Page 9, line 21. I presume you mean "only" not "on". Why is it that only GE mode is available?

Page 9, line 22. The stability here demonstrates that changes in mode differences are "indicative" not necessary for there to be calibration issues.

Page 10, line 37. So the dots here in Fig. 11 represent different frequencies, not just different months? I strongly suggest using different symbols for the different frequencies.

Page 11, line 27. Perhaps rephrase as "In many cases, the offset ... . Thus, changes in the reflectivity offset between the modes should be monitored, and used to identify periods where the calibration stability is suspect, and moving forward perhaps trigger more prompt additional external calibration evaluations".

Page 12, line 4. Seem redundant with the above comments on page 11.

---

## Author Comment (AC1) · 4 Jul 2019

See the attachment

Please also note the supplement to this comment:
https://www.atmos-meas-tech-discuss.net/amt-2019-34/amt-2019-34-AC1-supplement.pdf

---

## Author Comment (AC3) · 4 Jul 2019

**General comment:**

We would like to thank all three reviewers for their vey insightful comments. This study has been a humbling experience for the first author who has dedicated several years working on extracting scientific value out of the ARM facility radars and other sensors. A great challenge was to consolidate the differences between the ARM radars and generalized enough an algorithm initially developed by Alain Protat to work on a much larger dataset.

A project website has been developed and gives a graphical overview of the calibration procedure as applied to each site and radar system described in the manuscript. The web site is now available to the ARM radar user community. We hope to continue updating the material on the web site as the ARM program conducts additional field deployments. We also plan to expand our analysis to the European radar network.

http://doppler.somas.stonybrook.edu/CloudSat GlobalCalibrator/index.html

**Roger Marchand (Referee) rojmarch@uw.edu**

Received and published: 5 May 2019 Review of Manuscript amt-2019-34

Title: Calibration of the 2007-2017 record of ARM Cloud Radar Observations using CloudSat Authors: Kollias, Treserras and Protat

Overview:

I think publication of this technique and the results for ARM radars will be of value to many investigators and investigations that have (and will continue to) rely on ground-based radar datasets. While I was under no illusion that the ARM radars were well calibrated, I nonetheless found the results to be sobering.

Recommendation: Publish after minor revisions

General Comments:

1) A few more details on the technique. I largely follow the technique, but you need to add a few more details, see specific comments for Page 6. The goal should be to make it so that someone else could implement the approach given this description. In particular, please discuss uncertainties in estimated calibration corrections associated with equations 2 and 3, as well as the height range used to estimate the best offset.

We would like to thank the reviewer for his excellent comments that help us to clean up the methodology and provide additional information on how to reproduce our approach. Details comments are provided below

2) Differences in rules and thresholds for including or not including columns & Verification metrics. Do the different rules and thresholds for including or not including radar columns, which are to some degree necessarily different between CloudSat and ARM, matter? (see e.g. differences on Page 5, line 16; Page 7 line 17). I am concerned about the possibility that differences in the mean Z-profiles might be due simply to having different conditions or "distributions of cloudtypes" in each collection from which the mean-Zprofile is calculated. One way to check this would be to look not just at the mean-Z-profile but also to ask if the two profiles are based on a similar fraction of the observations in each set. Said another way, once you construct your "nonprecipitating CFAD" and pick your dBZethreshold (for calculating the mean-Z-profile), is the profile of cloud fraction associated with this dBZ-threshold the same for both CloudSat and ARM. If it is, then one can be confident that errors in the reflectivity correction due to differences in cloud populations will be small. I suggest creating a metric, such as the vertically integrated absolute cloud-fraction difference divided by the mean cloud fraction, and plotting this information along with the calibration corrections. Likewise, it would be interesting to see how this metric depends on the number of columns, which like figure #6, should give one a sense of what is a reasonable value for this quantity. Likewise, the cloud-top-height (CTH) distribution comparison you introduce (Fig 4d) provides confidence that the calibration correction is robust and that it is based on the same cloud populations. As far as I can see, after you introduce the idea of this as "a verification", you don't use it. At a minimum it seems like you should discuss whether the CTH distributions are consistently improved (made more similar) with the radar correction or not. Again, you might make a metric that expresses this improvement – though I suspect the above cloud fraction metric is likely better for this purpose.

The reviewer is correct. The cloud-top-height (CTH) distribution comparison are used in a qualitative way to verify that they converge as we get closer to the correct calibration offset. As part of the manuscript, we are also releasing a web site that provide graphics and animations for all the ARM sites and radar systems compared to CloudSat. You can clearly see that in CTH distribution comparison always get better at each site as we approach the RMSE. In few cases, the improvement in the CTH was not evident and the estimated calibration offsets were removed manually. We hope that the reviewer and the larger user community will find the material on the web site useful.

**http://doppler.somas.stonybrook.edu/CloudSat GlobalCalibrator/index.html**

3) Results for Darwin and the size of the analysis region I don't typically like to point to my own work when reviewing an article, but in this case I think some work that a former student of mine Zheng Liu, Tom Ackerman and I have done at Darwin is very germane to this study. Liu, Z., R. Marchand, and T. Ackerman (2010), A comparison of observations in the tropical western Pacific from groundâA× Rbased and satellite millimeterâ× A× Rwavelength× cloud radars, J. Geophys. Res., 115, D24206, doi:10.1029/2009JD013575. In particular, we compared CloudSat and the Darwin MMCR measurements and we investigated the size of the analysis region and sampling uncertainties in some detail. That study very much supports using a 300 km radius area and 6 month window. Note also figure 8 in this paper. While we did not derived a calibration offset, our results are broadly consistent with idea that ARM calibration was too LOW at Darwin in the 2006-7 (wet season), and agrees your with figure 9a (CloudSat – ARM difference of ~= 5 dB at this time).

We would like to thank the reviewer for pointing out to his previous work that is quite relevant to the results presented here. The suggested work is one of the first that compared ground-based and space-based millimeter wavelength radar observations. A reference was added in the revised manuscript regarding the finding of the study at Darwin.

Minor Comments:

Page 1, line 28. detail => detailed

The manuscript is revised according to your suggestion. Thank you.

Page 2, line 5. Opening sentence is awkward, rephrase. Perhaps "Part of the motivation for the ARM radar expansion, was to improve cloud microphysical retrievals through the use of dual-wavelength ratios, that is, making use of the relative difference in radar scattering at different wavelengths. This difference signal is often only a few dB and as one might expect, this .... "

This is a good way or rephrasing the opening sentence. The manuscript is revised according to your suggestion. Thank you.

Page 2, line 11. Do you mean calibrating ARM vertically pointing radars is more difficult than the WSR-88D network? What is being compared to what is not clear? Suggest you rephrase this and following sentence to be clearer and generally read better.

We agree with the reviewer. We rephrase the sentence as follows: "Soon after the National Aeronautics and Space Administration (NASA) Tropical Rainfall Measuring Mission (TRMM) spaceborne radar was in orbit, its remarkable stability made it a calibration standard and its comparison to the ground-based observations of the Weather Surveillance Radar -1998 Doppler (WSR-88D) network uncovered several issues with the calibration of the radars despite the mandate of the WSR-88D network on quantitative precipitation estimation and the implementation of routine calibration procedures [Bolen and Chandrasekar, 2000]. On the other hand, establishing routine calibration procedures based on engineering testing procedures or natural targets for the ARM profiling cloud radars is a far more challenging task."

Page 3, line 3. Change "... is such diverse" to "... to such a diverse set ...".

The manuscript is revised according to your suggestion. Thank you.

Page 3, line 31. "... on the same..." to "at the same".

The manuscript is revised according to your suggestion. Thank you.

Page 4: First paragraph: Mode analysis. How did you account for differences in the minimum detectable signal between the modes? I presume you only included neighboring time & range bins where both modes have a measurement with high SNR?

The reviewer is correct, we only used neighboring time/range bins with SNR >0. We have revised the text to make it more clear: "The difference [dB] in the measured radar reflectivity between two modes is estimated at heights where both modes provide observations (e.g., the MMCR mode 2 does not provide data below 3.6 km) with high Signal-to-Noise (SNR > 0 dB), and at ranges where the averaged profiles were correlated to filter our ranges where bid discrepancies due to radar artifacts were present" The last requirement (correlation of profiles in range) was introduced because the pulse compression can introduce range side lobes.

Page 4. The first paragraph launches into a discussion of mode differences (which is useful) but a bit confusing when one is expecting a comparison of CloudSat and ARM calibrations. I suggest breaking this paragraph about line 8 and adding. "Therefore as a prelude to comparing CloudSat and ARM, we begin with a comparison of reflectivity values between ARM radar modes. As will become clear later, changes in the reflectivities between modes is often, though not always, indicative of changes in overall calibration."

The manuscript is revised according to your suggestion. Thank you.

Page 4, line 13. I presume "bid" should be "big".

The manuscript is revised according to your suggestion. Thank you.

Perhaps Page 4, line 14. Specify period (6 months?).

The time window used to compare two ARM radar modes is 1-month since there is a large dataset to compare. The duration of the time window was added in the revised manuscript. Thank you.

Page 4, line 16. Perhaps change to read: "Overall, the mode reflectivity differences are small ( $\pm 2$  dB) and only occasionally are the differences much higher than 2 dB. While the absolute values of mode difference is in the next generation of ARM cloud profiling radars (KAZR and KAZR2) is often similar, arguably there are fewer jumps or rapid changes (except perhaps at OLI). In general it is difficult to identify which mode has a better calibration, because as will be shown, the calibration difference between CloudSat and ARM is typically larger than  $\pm 2$  dB.

**The manuscript is revised according to your suggestion. Thank you.**

\*\* Page 4: Second paragraph on difference between Protat and current approach. This paragraph is nearly impossible to follow if you don't already know what Protat 2010 did. In particular, I have no idea what ". . . a rigorous selection of the CloudSat overpasses within a certain radius to avoid any errors in the estimation of the proximity of CloudSat columns to the ARM site location" means. But other parts of this are confusing to me (and I am familiar with Protat 2010). I strongly encourage you to reorganize the manuscript such that you FIRST explain your approach in detail and ONLY at the end of this material highlight how this approach differs from Protat (2010).

The comment regarding the "... a rigorous selection of the CloudSat overpasses within a certain radius to avoid any errors in the estimation of the proximity of CloudSat columns to the ARM site location" refers to a software bug found in the original code provided to us by Alain and sometimes

was introducing CloudSat echoes that were not within the specified radius from the ARM site as being within the specified radius. We agree with the reviewer comment. We re-organized the description of the methodology, first we explain our approach in detail and at the end we highlight any differences to Protat 2010. We hope that the revised manuscript offers a clearer description of the methodology used in this study.

\*\* Page 6, equation 2. What is the justification for using a constant here? You used Rosenkranz and . . . ? Somehow you must have specified some set of atmospheric profiles to come up with these constants? Explain in enough detail so someone else could implement this idea. Nominally, I think it would have been better to calculate a set of gasses corrections for ARM (perhaps using ERA data just as CloudSat does). \*\*But in lieu of this, I think you need to address how much error (uncertainty) using this constant introduces in your calibration correction.

Initially, we wanted to use the ARM facility soundings to estimate the gaseous attenuation for the ARM radar observations. However, in most cases, the CloudSat overpasses and the ARM soundings which are available on 2-4 times per day are several hours apart. Our preliminary analysis indicated large differences in the two estimates especially during periods when large scale advection was present. Thus, we decided to use the same thermodynamic profile for gaseous attenuation for both the ARM and CloudSat observations. As the reviewer pointed out, CloudSat uses ERA which is available all the time along the CloudSat orbit. The use of the ERA-based CloudSat gaseous attenuation for the ARM radar eliminates any possible biases related to biases in the reanalysis and mitigates all uncertainty in the methodology used to estimate the conversion constant. The estimation of the conversion constant was performed using a large number of ARM soundings at each fixed or mobile site. The ARM soundings were used as input to Rosenkranz to estimate the  $G_{94-GHz}^{inv}$  and the ( $G_{35-GHz}$ ). The conversion factor was found to have a latitudinal dependency, thus, three different values we introduced. Per the reviewer request, we estimated the standard deviation of the parameter C and it was found to be around 0.5 for AWR, NSA, OLI and TMP and 0.8 for ENA, SGP and TWP. If we consider that the column integrated gaseous attenuation at 35-GHz is about 0.4 for AWR, NSA, OLI and TMP; 0.6 for ENA and SGP; and 1.0 for TWP, we can estimate the uncertainty introduced by using a constant C using Eq. 2. The uncertainty is: 0.14 dB for AWR, 0.09 dB for NSA, OLI and TMP, 0.15 dB for ENA and SGP and 0.2 dB for TWP. The following text was added in the methodology:

"The conversion factor is derived using [Rosenkranz, 1998] and a large number of ARM sounding and its average value depends on the ARM radar location (i.e.,  $1.45\pm0.5$  for AWR,  $2.08\pm0.5$  for NSA, OLI and TMP,  $3.36\pm0.5$  for ENA and SGP and  $4.03\pm0.5$  for TWP). Considering that the averaged integrated two-way attenuation at 35-GHz at these locations is 0.4 dB for AWR, NSA, OLI and TMP, 0.6 dB for ENA and SGP and 1.0 for TWP, the uncertainty introduced by using the conversion factor is 0.13 dB at AWR, 0.09 dB at NSA, OLI and TMP, 0.15 dB at ENA and SGP and 0.2 dB at the TWP sites."

\*\* Page 6, equation 3. What assumptions does this equation entail? Again, I think you need to address how this impacts the uncertainty of your calibration corrections?

Equation 3 is adopted by the methodology described in Protat (2010). In brief, Protat (2010) used a large database of in situ ice cloud microphysical measurements of the ice particle size distribution

gathered in different international field experiments, assumed a temperature-dependent massdimension relationship, and calculate radar reflectivity at 35 and 94-GHz using Mie theory. The scattering calculations results here fitted to extract the power-law relationship shown in Eq. 3. As the reviewer suggests, there are a lot of assumption in estimating this relationship.

Fortunately, in our analysis we use sensitive radars. Both CloudSat and the ARM profiling radar are capable of observing ice clouds with reflectivities as low as -30 dBZ throughout the upper troposphere. Only a very small fraction of the CloudSat and ARM observations used for the comparison exceed 0 dBZ. We examined the sensitivity of the calibration offsets to the application or not of the Eq 3 and it was found negligible. In any case, and for consistency with previous effort, we did apply Eq 3. The following text was added:

"Eq. 3 is based on assumption regarding the mas-diameter relationship of the ice particles used in the Mie scattering calculations. According to Eq. 3, differences in the radar reflectivity at 35- and 94-GHz start exceeding 1-dB at about 0 dBZ at 35-GHz. In the analysis presented here, the vast majority of the 35-GHz radar ice reflectivities used are below 0 dBZ. Thus, any uncertainty introduced by using Eq. 3 is considered negligible."

\*\* Page 6, line 17. I am not sure I understand this definition, there seems to be a grammatical error here. Do you simply mean "precipitating column = 10% (or more) of the radar volumes below the FL have ANY reflectivity (even if it is -30 dBZe). So if FL is 2 km at there are 20 bins below 2km, if 3 bins have ANY detection (even if it is just a low cloud) you are calling it precipitating?

The reviewer is correct, there is a radar reflectivity threshold that is missing. The correct phrase is: "An ARM column is considered to be precipitating if at least 10% of the range gates below the freezing level report echoes higher than -10 dBZ". We have added this correction to the revised manuscript.

What do you do if the FL is near the surface (in the CloudSat clutter) or with only a few ARM bins  $\sim 5$ ?

The same rules apply. However, our FL estimate is always conservative (always higher by at least 500 m than the actual FL) and we never use CloudSat observations in the lowest 1.5 km of the atmosphere

Why do you use a different threshold 35% for CloudSat? This seems arbitrary.

The following text was added in the revised manuscript: "The threshold selection 35% for CloudSat is based on an extensive sensitivity study. In particular, we estimated the sensitivity of calibration offset for different allowed % (from 0 to 100%) of CloudSat echoes with radar reflectivity exceed -10 dBZ below the freezing level. The calibration offset exhibited systematic biases for threshold values higher than 35%. Thus, the threshold value of 35% was selected to maximize the number of CloudSat columns used and at the same time eliminate the possibility of systematic biases."

\*\* Page 6, items 2 to 4. This material seems important and needs to be better explained. In particular, how does the "degradation" work? I presume you mean that the mean-Z-profiles, are obtained from the CFADS in step 4 by summing bins with dBZ > Threshold > Minimum Detectable Signal (MDS) weighted by the bin dBZe? (I note without weighting this just give you the "profile of cloud fraction"). If yes, it might be important to choose threshold that is + 3 to 5 dB larger than the MDS.

The CloudSat only echoes with cloud mask values higher than 20 are used. Based on these filtered values, we estimate the CloudSat MSD (one constant value for the entire troposphere). In ARM, we are doing pretty much what the reviewer suggests, we use a threshold of +3 to +5 dB larger that the theoretical MDS. This relates to the next question of the reviewer.

Where/How does the SNR > -15 mentioned early come in?

Profiling cloud radars utilize high pulse repetition frequency (3-10 kHz) and long averaging (1-2 sec). Thus, are capable at extracting signal at low SNR conditions. Not all collected samples are independent (this depends on the signal bandwidth, or otherwise the Doppler spectrum width). For typical conditions of spectrum width encountered in weak reflectivity targets, we estimate that we have 1000 or more independent samples used in the moment estimation. The resulting reduction in the received noise variance is given by the square root of the number of samples  $\sim 15$  dB. Although our experience with working with the ARM facility profiling cloud radars is that signal extraction is possible at even lower that -15 dB SNR conditions, we decided to accept only radar reflectivity estimates with SNR>-15 to ensure the removal of any outliers.

Page 7. It seems you address the issues of the number of columns in detail later in the text, but do NOT the distance issue. (see also general comment #3). Perhaps add some discussion and/or better yet show result for OLI site (where you have lots of data) – add a line to fig 7 – for results based on 100 vs. 300 km?

A details sensitivity study was performed at each ARM radar site to investigate the impact of the selected maximum distance for CloudSat observations on the estimated calibration offsets. It was found that a distance of 200 km was optimum for most locations. The following text was added in the revised manuscript: "In particular, we examined the sensitivity of the estimated calibration offset to the selected maximum distance of the CloudSat observations. Using difference distance values from 100 to 300 km every 25 km at different sites, we investigated the behavior of the estimated calibration offsets. Our analysis indicated that a maximum distance of 200 km was optimum for most ARM locations and therefore, was selected as a fixed value throughout the study."

\*\* Page 7/8, analysis on number of columns vs. number of good columns ? I like very much the analysis you have included on the number of columns. But unless I misunderstand you are counting ALL columns here. Not the number of good columns (i.e. columns which are devoid of high/ice clouds or precipitating). I think it would be far more sensible to count the number of columns with good data (and set a minimum threshold on this) rather than all radar columns.

The number of columns reported is the number of good columns given the selection criteria described in the methodology section. This is consistent with Figure 5 that shows that the number of samples strongly depends on seasonal cloud variability.

Page 9, line 21. I presume you mean "only" not "on". Why is it that only GE mode is available?

The reviewer is correct, we do mean only. Regarding the comment that only GE mode is available: After checking with the ARM facility, we discovered that there is a chirp mode that was given a different filename in the archive and thus, it is not possible for us to find it.

Page 9, line 22. The stability here demonstrates that changes in mode differences are "indicative" not necessary for there to be calibration issues.

The reviewer is correct. We have indicated in the revised summary and in the text that changes in model differences are only indicative of time periods with calibration issues.

Page 10, line 37. So the dots here in Fig. 11 represent different frequencies, not just different months? I strongly suggest using different symbols for the different frequencies.

In the revised manuscript, Fig 11 has been revised according to your suggestion. Thank you.

Page 11, line 27. Perhaps rephrase as "In many cases, the offset .... Thus, changes in the reflectivity offset between the modes should be monitored, and used to identify periods where the calibration stability is suspect, and moving forward perhaps trigger more prompt additional external calibration evaluations".

The manuscript is revised according to your suggestion. Thank you.

Page 12, line 4. Seem redundant with the above comments on page 11.

The summary has been revised and we believe any redundancy has been removed.

---

## Author Response (AR1)

**Genera comment:**

We would like to thank all three reviewers for their vey insightful comments. This study has been a humbling experience for the first author who has dedicated several years working on extracting scientific value out of the ARM facility radars and other sensors. A great challenge was to consolidate the differences between the ARM radars and generalized enough an algorithm initially developed by Alain Protat to work on a much larger dataset.

A project website has been developed and gives a graphical overview of the calibration procedure as applied to each site and radar system described in the manuscript. The web site is now available to the ARM radar user community. We hope to continue updating the material on the web site as the ARM program conducts additional field deployments. We also plan to expand our analysis to the European radar network.

**http://doppler.somas.stonybrook.edu/CloudSat GlobalCalibrator/index.html**

**Anonymous Referee #1**

Received and published: 10 April 2019 Review of the article titled "Calibration of the 2007-2017 record of ARM cloud radar observations using CloudSat" by Kollias and coauthors for publication in AMT.

The authors have compared the reflectivity from vertically pointing Doppler cloud radars at the ARM sites to the reflectivity from radar onboard polar orbiting satellite. The goal of this study is to characterize the performance of the ground-based radars, as the spaceborne radar is well-calibrated. They find significant calibration offset for the ground-based radars throughout the 10-year period, and inherent inconsistencies between the different modes of them. The technique is already well-established and used here in a relatively straightforward manner. The article is overall okay but needs several small tweaks in writing. Due the number of small corrections listed below; I recommend this article for major revisions.

**Major Suggestions:**

1) As the authors have already made CFAD of all the ARM radars, it will be relatively straightforward for them to calculate the minimum detectable signal (MDS) for them. You can just pick up the bottom 5% of reflectivity at 1 km and make its average. This will greatly assist the scientific user community, as it is unclear how sensitive are the ARM radars and if their sensitivity has changed through years. You already have the data for calculating this and hence will be a worthy and useful effort. If you do this, then you can add this as another column in Table 1. Thanks.

We would like to thank the reviewer for his comment. The detail tables with the calibration offsets, the RMSE and the number of samples have been provided to the ARM infrastructure. This study is not an official reference document for the ARM facility and the author team is not representing ARM in any capacity. Our understanding is that the ARM facility will consider our findings and decide a path forward on how to report them to the user community. This includes the requested

minimum detectable signal information that as the reviewer correctly suggests can be relatively easy released by ARM.

2) Please add one more column to Table 1 and report the average and standard deviation of the calibration offset for each radar. This table will be very useful for users who'd like to use your calibration offsets in their research. Please add the different modes of KAZR and MMCR in the rows. I understand that this will be an average through many years, but still worthy of reporting. Thanks.

If it fair to state that a considerable amount of work and analysis was performed to complete this long-term ARM – CloudSat calibration. As part of the manuscript, we are also releasing a web site that provide graphics and animations for all the ARM sites and radar systems compared to CloudSat. We hope that the reviewer and the larger user community will find the material on the web site useful.

**http://doppler.somas.stonybrook.edu/CloudSat GlobalCalibrator/index.html**

3) The article ends abruptly, and you only provide a brief summary without discussing the implications of your results. It will be good if you can devote one paragraph each on the following two things i) the impact of calibration offset on the regular data products produced by the ARM program. I did a quick search and the radar reflectivity is used for doing microphysical retrievals like MicroBase and cloud drop concentrations. Please discuss how a calibration offset might affect these data-products. ii) The lead author has significant expertise in retrieving vertical air motion and microphysical properties from ground-based radars. A quick search made me realize that scientists have also used radar reflectivity in those studies in addition to using mean Doppler. It will be good if you can elaborate on how your results will impact the results previous studies by you and from Giangrande, Verlinde, Luke, Shupe, Dong, Chiu etc. So please add two separate paragraphs at the end and rename the section as "Summary and Discussion".

The reviewer is correct, the summary does ends abruptly. The impact of the calibration offsets reported here is not negligible and as the reviewer suggested, will affect any retrievals and/or data product that depends on calibrated radar reflectivity values. Here is the text added in the revised manuscript: "In most cases, the observed calibration offsets exceeded this uncertainty value suggesting that the ARM profiling radar record contains considerable calibration biases. The reported calibration biases are expected to have a large impact on routine ARM microphysical data products such as the Continuous Baseline Microphysical Retrieval (MICROBASE) value-added product [Zhao et al., 2012]. In addition, cloud retrieval techniques and associated products are impacted by the reported calibration offsets ([Shupe et al., 2015]; [Dong et al., 2014]). For reference, a 3-dB calibration offset is equivalent to a factor of 2 bias in hydrometeor content or number concentration retrievals. The estimated calibration offsets, the RSME's and the number of samples as a function of time for each radar system evaluated here have been provided to the ARM facility. The ARM facility is currently considering reprocessing of the ARM radar record with these new calibration offsets."

Minor Issues:

It is unclear to me if the authors are referring to the funding agency ARM or their observatories through the user facility. I recommend using the ARM Climate User Facility throughout the article. Thanks.

The reviewer is correct. ARM has not been a program since it was designated a user facility. Thus, we use "ARM facility" throughout the manuscript.

Page-1, Line 16: Add "collectively" before "Over". Thanks.

The manuscript is revised according to your suggestion. Thank you.

Page-1, Line 12: Add :1990s

The manuscript is revised according to your suggestion. Thank you.

Page-1, Line 15: the sentence doesn't read well, please rephrase.

The sentence is revised as follows: Here, a well-characterized spaceborne 94-GHz cloud profiling radar (CloudSat) is used to characterize the calibration of the ARM cloud radars. The calibration extends from 2007 to 2017 and includes both fixed and mobile deployments. Thank you.

Page 2, Line 8: "Surprise" not "surprised".

The manuscript is revised according to your suggestion. Thank you.

Page 3: Add outline of the paper before the section 2.

The manuscript is revised according to your suggestion. Thank you. Here is the revised last paragraph of the introduction:

"In section 2, the ARM facility cloud radars are presented and the Protat et al. [2011] methodology is revised and improved. Section 3 presents the results from the application of the calibration procedure to almost the entire record of ARM profiling cloud radar observations at the fixed and mobile sites from 2007 to the end of 2017 (at total of 43.5 years of radar observations). Finally, section 4 presents a summary on our finding and their implications. The application of the technique is such diverse set of radar systems and locations is expected to demonstrate the applicability of this approach to existing profiling radar networks such as the ARM facility and the future European research infrastructure network for the observations of Aerosol, Clouds and Trace gases (ACTRIS)."

Page 3, Line 19: "At couple of sites".

The manuscript is revised according to your suggestion. Thank you.

Page 4, line 1-2: Please rephrase and remove "us".

The phrase is removed since it is basically a repetition of what is already mentioned in the previous paragraph. Thank you.

Page 4, Line 15: "Computed" and not "computer".

The manuscript is revised according to your suggestion. Thank you.

Page 5 Line 10 and Page 7 line 13: It is unclear which numbers to believe.

The correct range is 200 to 300 km. We corrected the inconsistency between the numbers reported in page 5 and page 7 regarding the maximum distance of CloudSat observations used in the calibration. Thank you.

Page 5-6: It will be good if you mention the equation used for doing gaseous correction in CloudSat. Thanks.

The gaseous attenuation correction in the operational CloudSat products (R04/R05) is based on the Millimeter-wave Propagation Model (MPM) of Liebe 1989

H. J. Liebe, "MPM—An atmospheric millimeter-wave propagation model," Int. J. Infrared Millim. Waves, vol. 10, no. 6, pp. 631–650, Jun. 1989

This information was added in the revised manuscript. Thank you.

Page 8 Line 15: "were" not "where".

The manuscript is revised according to your suggestion. Thank you.

Page 10 Line 15: you mean "observatory" and not "laboratory"?

The manuscript is revised according to your suggestion. Thank you.

Page 11 Line 20: there is a typo, it should be 616 samples according to Page 7 line 32. Thanks.

The reviewer is correct. In the summary, we refer to the total number of 6-months ARM-CloudSat comparisons. A phrase was added to clarify the number of cases we were able to estimate a calibration coefficient. The revised manuscript reads as follows: "A total of 653 ARM – CloudSat comparisons are performed using a running 6-month time window. Acceptable calibration coefficients are estimated in 616 of the cases, a 94.3% success rate". Thank you.

**Anonymous Referee #2**

Received and published: 2 May 2019

The present manuscript describes the efforts of the Authors to calibrate a long series of groundbased radar measurements using space-borne radar measurements from CloudSat. This task is all the more important as it can affect the quality of atmospheric retrievals. Moreover, the calibration of such a long time series on a common ground helps greatly the study of the climate on such time scales. The article provides in-depth information into the operation and maintenance of the ARM radar network. As such, it makes publicly available information that otherwise would be known only to the few expert users/members of the ARM program. For that alone, this manuscript is worth publishing.

The Authors follow a clear path to describe their datasets, its quality control, and the methods to collocate and optimize the calibration assessment. Various graphs provide a nice illustration of the performance of the proposed method. Before publishing this article, these are the points I would like the Authors to address:

1. The article needs a serious editorial revision to correct for grammar errors and typos. In particular, the Authors should revise the tenses of the verbs for consistency.

We would like to thank the reviewer about his numerous comments regarding grammar errors and typos in the original manuscript. All the reviewer suggestions and those from the other two reviewers have been included in the revised manuscript. Furthermore, we revise the tense of the verbs when needed for consistency.

2. Please provide a table listing the various acronyms, and please define these acronyms in the article at their first occurrence.

All the acronyms in the text are now defined at their first occurrence.

3. As a general question, would the statistical method that you use (to match the mean profiles) work if you also match the envelope of the CFAD (lower and upper quantiles)? This envelope may have useful information, e.g. on the variability of the reflectivity profile over time or space.

We believe that the statistical method used here (RMSE of the mean profiles) will also work on other CFAD parameters such as the lower and upper quantiles.

4. Would the Authors see any merit/advantage in applying their optimal calibration method to other satellite datasets collocated with ARM radars? Could you please comment on this in your article?

This is an interesting suggestion. Over the recent years, the ARM facility has acquired and operated several cm-wavelength radars that are also in need of calibration. One could see that observations from NASA's GPM Dual-Frequency Precipitation Radar could be used to evaluate the calibration of the ARM facility cm-wavelength radars. In addition to referencing radars, we believe there is

great value in evaluating ARM observations against those provided by geostationary satellites (GOES-R, MSG) and the A-train. The following text was added in the revised manuscript at the end of the summary and discussion section:

"Furthermore, there is merit in extending the presented analysis to other satellite measurements. For example, NASA's Global Precipitation Mission (GPM) Dual-Frequency Precipitation Radar (DPR) observations could be used in a similar manner to evaluate the calibration of the ARM facility cm-wavelength radars [Lamer et al., 2019]. In addition to radar calibration, the statistical comparison between cloud and precipitation properties such as cloud base height, cloud thickness, precipitation occurrence and intensity and liquid water path measured at the ARM facility and those derived by research satellites such as NASA'S A-train constellation [Stephens et al., 2018] should be considered. The ARM facility provides a bottom-up view of clouds and precipitation with superior vertical resolution especially in the boundary layer. Statistically significant differences with the top-down view provided by the A-train satellites should be considered when conducting cloud-scale process studies using global satellite datasets."

Please also note the supplement to this comment: https://www.atmos-meas-tech-discuss.net/amt-2019-34/amt-2019-34-RC2- supplement.pdf

Thank you very much for annotated supplement!

**Following the reviewer's recommendation Fig. 4 was updated**

**Roger Marchand (Referee) rojmarch@uw.edu**

Received and published: 5 May 2019 Review of Manuscript amt-2019-34

Title: Calibration of the 2007-2017 record of ARM Cloud Radar Observations using CloudSat Authors: Kollias, Treserras and Protat

Overview:

I think publication of this technique and the results for ARM radars will be of value to many investigators and investigations that have (and will continue to) rely on ground-based radar datasets. While I was under no illusion that the ARM radars were well calibrated, I nonetheless found the results to be sobering.

Recommendation: Publish after minor revisions

General Comments:

1) A few more details on the technique. I largely follow the technique, but you need to add a few more details, see specific comments for Page 6. The goal should be to make it so that someone else could implement the approach given this description. In particular, please discuss uncertainties in estimated calibration corrections associated with equations 2 and 3, as well as the height range used to estimate the best offset.

**We would like to thank the reviewer for his excellent comments that help us to clean up the methodology and provide additional information on how to reproduce our approach. Details comments are provided below**

2) Differences in rules and thresholds for including or not including columns & Verification metrics. Do the different rules and thresholds for including or not including radar columns, which are to some degree necessarily different between CloudSat and ARM, matter? (see e.g. differences on Page 5, line 16; Page 7 line 17). I am concerned about the possibility that differences in the mean Z-profiles might be due simply to having different conditions or "distributions of cloudtypes" in each collection from which the mean-Zprofile is calculated. One way to check this would be to look not just at the mean-Z-profile but also to ask if the two profiles are based on a similar fraction of the observations in each set. Said another way, once you construct your "nonprecipitating CFAD" and pick your dBZethreshold (for calculating the mean-Z-profile), is the profile of cloud fraction associated with this dBZ-threshold the same for both CloudSat and ARM. If it is, then one can be confident that errors in the reflectivity correction due to differences in cloud populations will be small. I suggest creating a metric, such as the vertically integrated absolute cloud-fraction difference divided by the mean cloud fraction, and plotting this information along with the calibration corrections. Likewise, it would be interesting to see how this metric depends on the number of columns, which like figure #6, should give one a sense of what is a reasonable value for this quantity. Likewise, the cloud-top-height (CTH) distribution comparison you introduce (Fig 4d) provides confidence that the calibration correction is robust and that it is based on the same cloud populations. As far as I can see, after you introduce the idea of this as "a

verification", you don't use it. At a minimum it seems like you should discuss whether the CTH distributions are consistently improved (made more similar) with the radar correction or not. Again, you might make a metric that expresses this improvement – though I suspect the above cloud fraction metric is likely better for this purpose.

The reviewer is correct. The cloud-top-height (CTH) distribution comparison are used in a qualitative way to verify that they converge as we get closer to the correct calibration offset. As part of the manuscript, we are also releasing a web site that provide graphics and animations for all the ARM sites and radar systems compared to CloudSat. You can clearly see that in CTH distribution comparison always get better at each site as we approach the RMSE. In few cases, the improvement in the CTH was not evident and the estimated calibration offsets were removed manually. We hope that the reviewer and the larger user community will find the material on the web site useful.

**http://doppler.somas.stonybrook.edu/CloudSat GlobalCalibrator/index.html**

3) Results for Darwin and the size of the analysis region I don't typically like to point to my own work when reviewing an article, but in this case I think some work that a former student of mine Zheng Liu, Tom Ackerman and I have done at Dar